# RELIABLE TEST-TIME ADAPTATION VIA AGREEMENT-ON-THE-LINE

## ABSTRACT

Test-time adaptation (TTA) methods aim to improve robustness to distribution shifts by adapting models using unlabeled data from the shifted test distribution. However, there remain unresolved challenges that undermine the reliability of TTA, which include difficulties in evaluating TTA performance, miscalibration after TTA, and unreliable hyperparameter tuning for adaptation. In this work, we make a notable and surprising observation that TTAed models strongly show the agreement-on-the-line phenomenon (Baek et al., 2022) across a wide range of distribution shifts. We find such linear trends occur consistently in a wide range of models adapted with various hyperparameters, and persist in distributions where the phenomenon fails to hold in vanilla model (*i.e.*, before adaptation). We leverage these observations to make TTA methods more reliable from three perspectives: (i) estimating OOD accuracy (without labeled data) to determine when TTA helps and when it hurts, (ii) calibrating TTAed models again without any labeled data, and (iii) reliably determining hyperparameters for TTA without any labeled validation data. Through extensive experiments, we demonstrate that various TTA methods can be precisely evaluated, both in terms of their improvements and degradations. Moreover, our proposed methods on *unsupervised* calibration and hyperparameters tuning for TTA achieve results close to the ones assuming access to ground-truth labels, in both OOD accuracy and calibration error.

## 1 INTRODUCTION

Machine learning models often fail to generalize to new distributions (Arjovsky et al., 2020; Gulrajani & Lopez-Paz, 2021; Sagawa et al., 2022) – so-called out-of-distribution (OOD) data — which differ from the one they were trained on, referred to as in-distribution (ID) data. This can lead to a significant degradation in their performance during test time. Recently, there has been a surge in research on test-time adaptation (TTA), a technique that adapts models to the target distribution using only unlabeled test data. These involve adaptation strategies including estimating test-time feature statistics (Schneider et al., 2020), self-supervision (Sun et al., 2020; Liu et al., 2021; Gandelsaman et al., 2022), entropy minimization (Liang et al., 2020; Wang et al., 2021; Zhang et al., 2022; Niu et al., 2022; 2023), and self-training with pseudo-labels (Liang et al., 2020; Rusak et al., 2022; Goyal et al., 2022). These efforts have aimed to enhance model robustness in the face of distribution shifts where labeled data is unavailable.

Despite the progress in TTA, several critical bottlenecks persist, undermining the reliable applications of such adaptation methods in practice. Firstly, TTA is not universally effective for all distribution shifts and can sometimes lead to performance degradation (Wang et al., 2021; Liu et al., 2021; Zhao et al., 2023). Moreover, the absence of labeled test data hinders the evaluation of model performance in practice, thereby making it unclear in advance whether these methods will work or not. Secondly, TTA methods often result in poorly calibrated models (Eastwood et al., 2022; Chen et al., 2022; Rusak et al., 2022), posing potential risks in safety-critical applications. Thirdly, TTA methods are often extremely sensitive to their hyperparameters during adaptation (Boudiaf et al., 2022; Zhao et al., 2023), and their tuning procedures lack clarity. Most of them often follow the same settings of the previous studies (Niu et al., 2022; 2023), or rely on some held-out labeled data (Zhang et al., 2022; Iwasawa & Matsuo, 2021; Goyal et al., 2022), which might be unavailable in practice. To our knowledge, there is little work analyzing how to effectively address these shortcomings.

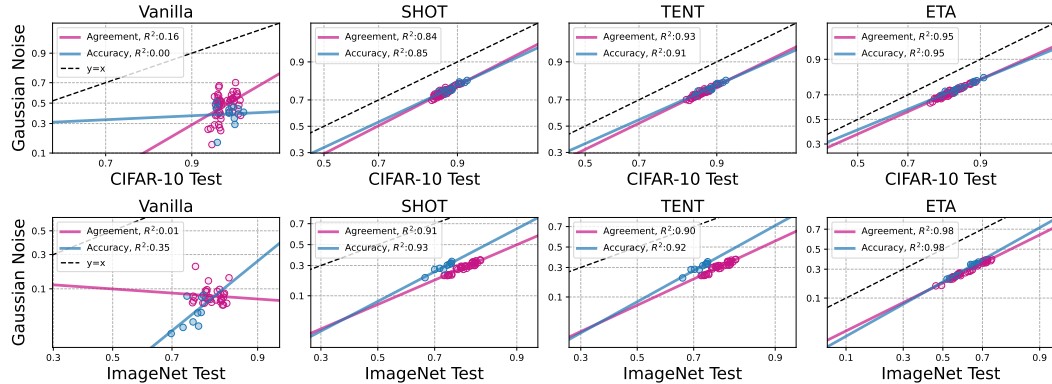

Figure 1: Accuracy (%) (blue dots) and agreement (%) (pink dots) of the TTAed models (different architectures) tested on ID (x-axis) and OOD (y-axis). Each blue and pink line denotes the linear fit of the blue and pink dots, respectively. The axes are probit scaled. Various TTA methods make the correlations in agreement and accuracy notably stronger than their base counterparts, with their $R^2$ values substantially increased. We test SHOT, TENT, and ETA on CIFAR10 (first row) and ImageNet (second row) against their Gaussian Noise corruption. Additional results in appendix A.6.

In a separate line of work, Baek et al. (2022) show that the OOD agreement between classifiers (*i.e.*, the average extent to which two classifiers make the same prediction on an unlabeled datasets) shows a strong linear correlation with their ID agreement, akin to the same phenomenon showed for ID versus OOD *accuracy*, demonstrated by Miller et al. (2021). Taken together, these phenomena, the so-called agreement-on-the-line (AGL) and accuracy-on-the-line (ACL) present a method for assessing OOD accuracy without labeled data.

In this paper, we observe a noteworthy phenomenon: after applying TTA, AGL and ACL *persist* or even hold to a *stronger* degree than in their base counterparts. In other words, when we assess the accuracies and agreements of the models adapted to OOD data, the strong correlations in ID vs. OOD consistently hold across distribution shifts, including those where vanilla models do not exhibit such trends. Interestingly, these correlations occur not only when TTA improves OOD accuracy, but also when it fails to enhance or even negatively affects OOD accuracy, especially under real-world shifts. We also observe such trends among the adapted models with varying values of their adaptation hyperparameters, including learning rates, the number of adaptation steps, and others.

These findings, with strong AGL and ACL after TTA, lead to the enhancement of TTA methods for improved reliability. We can first *predict* the effectiveness of TTA methods, *i.e.*, whether they succeed or fail and to what extent, across distribution shifts. Specifically, our approach uses the ALine-S and ALine-D techniques from Baek et al. (2022), and applies them to test-time adapted models. The result is that, without any labeled data at all, we can estimate the accuracy of TTAed models better than we can for vanilla models, especially for shifts where vanilla falls short (estimation error of 14.22% in vanilla vs. 1.51% after TENT (Wang et al., 2021) on CIFAR10-C Gaussian Noise). Such estimation results also enable the *identification* of shifts where TTA methods might potentially struggle to improve accuracy, such as in ImageNetV2 (Recht et al., 2019). Second, we introduce a novel variant of the temperature-scaling method, which achieves model calibration solely through estimated accuracy, representing an unsupervised approach that eliminates the need for labeled data as required by the original temperature scaling (Guo et al., 2017). We observe that it effectively reduces the expected calibration error (ECE) (Guo et al., 2017) close to the best achievable lower-bound using ground-truth labels. Finally, we introduce the *reliable hyperparameter optimization* strategy for adaptations without access to labels; selecting model with the highest ID accuracy. Across all TTA baselines we employ, the majority of models chosen through our approach exhibit performance comparable to those selected using ground-truth labels, resulting in an accuracy gap less than 1% on CIFAR10-C.

To summarize our contributions:
- We observe that AGL and ACL trends between TTAed models persist or can be stronger than those before adaptation, and such trends robustly occur across models with different architectures or TTA setups, on both synthetic and real-world shift datasets.

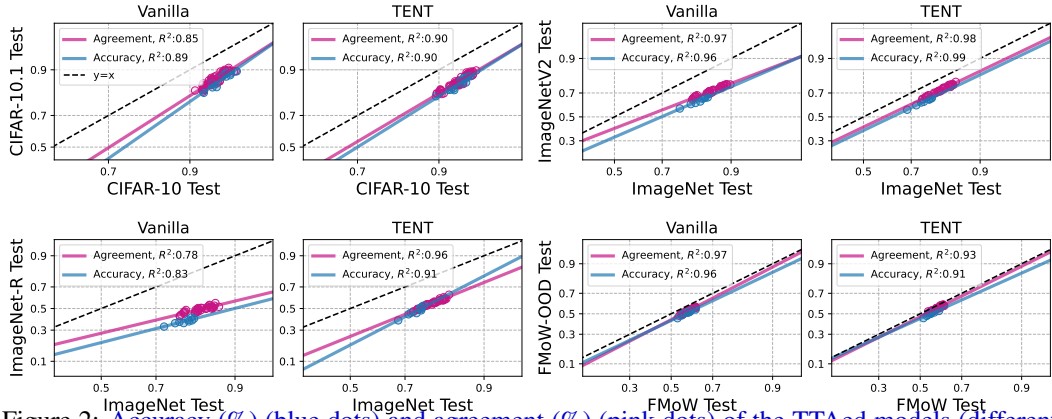

Figure 2: Accuracy (%) (blue dots) and agreement (%) (pink dots) of the TTAed models (different architectures) tested on ID (x-axis) and OOD (y-axis) test set. Each blue and pink line denotes the linear fit of the blue and pink dots, respectively. The axes are probit scaled. Across real-world shifts, such as CIFAR10.1, ImageNetV2, ImageNet-R, and FMoW-WILDS, the models after TTA maintain their strong linear trends. Notably, this observation also holds true despite possible accuracy degradation, e.g. on CIFAR10.1 and ImageNetV2 (See Table 2).

- By leveraging this phenomenon in TTA, we enhance TTA methods in OOD accuracy predictions, calibration, and reliable hyperparameter tuning—all without the need for labels. Our methods excel in broader distribution shifts compared to their base counterparts in accuracy estimation, and also demonstrate OOD accuracy and calibration performance comparable to those based on labeled data.

## 2 STRONG AGREEMENT-ON-THE-LINE AFTER TTA

This section presents the motivating finding of the paper, an empirical study highlighting the fact that TTAed models often exhibit better AGL than their vanilla counterparts.

### 2.1 EXPERIMENTAL SETUP

**Datasets and models.** We evaluate on both synthetic corruptions (CIFAR10-C, CIFAR100-C, ImageNet-C (Hendrycks & Dietterich, 2019)) with highest severity, datasets reproductions (CIFAR10.1 (Recht et al., 2018), CIFAR10.2 (Lu et al., 2020), ImageNetV2 (Recht et al., 2019)), and real-world shifts (ImageNet-R (Hendrycks et al., 2021), FMoW-WILDS (Christie et al., 2018; Sagawa et al., 2022)). We leverage a variety of different network architectures, which encompass ResNet (He et al., 2016; Zagoruyko & Komodakis, 2017), ResNext (Xie et al., 2016), VGG (Simonyan & Zisserman, 2015), GoogLeNet (Szegedy et al., 2014), DenseNet (Huang et al., 2017), and MobileNet (Sandler et al., 2018) with differing depths and widths. For evaluation on ImageNet and its shifts, we use pretrained weights publicly accessible from torchvision[1], except for TTT and ConjPL that require additional training procedures, *i.e.*, self-supervision or training with polyloss (Leng et al., 2022). Otherwise, we train models ourselves with source data. For additional specifications on models, see appendix A.1.

**Test-time adaptation baselines.** To gain generality, we test TTA methods that involve different update parameters (*e.g.*, batch normalization (BN), entire encoder parameters), objectives (*e.g.*, entropy minimization, self-supervision task), and source-training objectives (*e.g.*, cross-entropy loss, polyloss (Leng et al., 2022)). These include BN_Adapt (Schneider et al., 2020), SHOT (Liang et al., 2020), TTT (Sun et al., 2020), TENT (Wang et al., 2021), ConjPL (Goyal et al., 2022), ETA (Niu et al., 2022), and SAR (Niu et al., 2023). We examine their key adaptation hyperparameters that are shared among all baselines, including learning rates, number of adapt steps, and batch size. We also test different checkpoints of the source-trained model as another possible hyperparameter to select.

---

[1]https://pytorch.org/vision/stable/models.html

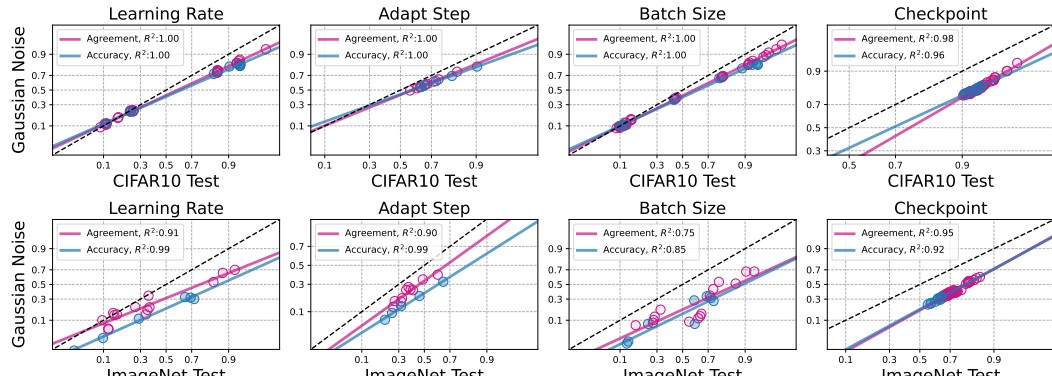

Figure 3: Accuracy (%) (blue dots) and agreement (%) (pink dots) of the TTAed models (different hyperparameters) tested on ID (x-axis) and OOD (y-axis) test set. Each blue and pink line denotes the linear fit of the blue and pink dots, respectively. The axes are probit scaled. We observe strong AGL and ACL among models adapted with varying values of TTA setups, including learning rates, the number of adapt steps, batch sizes, and checkpoints of the source-trained model. We plot the results of TENT-adapted models on CIFAR10-C Gaussian Noise in first row, and ETA-adapted models on ImageNet-C Gaussian noise in second row.

**Calculating agreement.** Given any pair of models $(h, h') \in \mathcal{H}$ that are tested on distribution $\mathcal{D}$, the expected agreement of the models are defined as

$$\text{Agreement}(h, h') = \mathop{\mathbb{E}}_{x \sim \mathcal{D}} \left[ \mathbb{1}\{h(x) = h'(x)\} \right], \tag{1}$$

where $h(x)$ denotes the most probable class of the model $h$ given data $x$. Following Miller et al. (2021) and Baek et al. (2022), we apply the inverse of the cumulative density function of the standard Gaussian distribution, namely probit-transformation ($\Phi^{-1} : [0, 1] \to [-\infty, \infty]$), on the axes of accuracy and agreement, for a better linear fit.

## 2.2 MAIN OBSERVATION

**Agreement (and accuracy) on-the-line persists or holds stronger after TTA.** We find that after TTA methods, there are strong correlations in agreement (and accuracy) between ID and OOD across both synthetic and real-world shifts. Most importantly, we notice that such trends become more persistent after adaptations, exhibiting strong correlations even across datasets where the models before adaptations have failed to have, as noted in Miller et al. (2021) and Baek et al. (2022).

We show such phenomenon in Figure 1: when tested on CIFAR10 test vs. Gaussian-Noise corruption, models with different architectures have weak correlations with their coefficients of determination ($R^2$) values being significantly low. After applying various TTA methods, SHOT, TENT, ETA, and SAR, these models consistently have much stronger AGL and ACL (*e.g.*, $R^2$ improves $0.16 \to 0.95, 0.00 \to 0.95$ after ETA), as well as the alignments between the lines. Furthermore, these trends also occur in real-world shift datasets, where TTA sometimes even fails to improve OOD accuracy, as shown in Figure 2. Here, we examine on CIFAR10.1, ImageNetV2, and ImageNet-R, and FMoW-WILDS, where models before and after TENT have similar linear trends and maintain high $R^2$ values. In particular, as seen in the plots of CIFAR10.1 and ImageNetV2, TENT does not improve generalization, or even results in degradation (Wang et al., 2021; Zhao et al., 2023). This highlights the consistent AGL and ACL trends across TTA methods, irrespective of TTA's actual performance improvement across diverse distribution shifts where it may succeed or falter.

**Such linear trends are also observed when varying TTA hyperparameters.** We find that these trends in TTA can also be obtained by leveraging models adapted with varying values of certain hyperparameters. As mentioned in Section 2.1, we examine learning rates, the number of adaptation steps, batch size, and the checkpoints of the source-trained model.

Here we fix the model architecture while systematically varying specific hyperparameters within defined ranges (See appendix A.1). Figure 3 shows that models adapted with different hyperparameter values exhibit strong AGL trends among them, with their $R^2$ values close to 1, when tested on

---

**Algorithm 1** Accurate Estimation of TTA performance

---

1: **Input:** Labeled ID data $\mathcal{X}_{\text{ID}}, \mathcal{Y}_{\text{ID}}$, unlabeled OOD data $\mathcal{X}_{\text{ID}}$, a set of ID-trained $n$ models $\mathcal{H} = \{h_\theta, ..., h_\theta\}$, sets $\mathcal{P}_{\text{ID}}, \mathcal{P}_{\text{OOD}}$.
2: **Algorithms:** TTA objective $\mathcal{L}_{\text{TTA}}(\cdot)$, ALine-S/D$(\cdot)$.
3: ───────────────────────────────────────────────
4: Initialize $\mathcal{P}_{\text{ID}} = \emptyset, \mathcal{P}_{\text{OOD}} = \emptyset$
5: **for** batch $x_{\text{ID}}, x_{\text{OOD}}$ in $\mathcal{X}_{\text{ID}}, \mathcal{X}_{\text{OOD}}$ **do**
6:     **for** $h_\theta \in \mathcal{H}$ **do**
7:         $\theta \leftarrow \theta - \eta \nabla \mathcal{L}_{\text{TTA}}(h_\theta(x_{\text{OOD}}))$                                    ▷ Apply TTA
8:         $\mathcal{P}_{\text{ID}} = \mathcal{P}_{\text{ID}} \cup h_\theta(x_{\text{ID}})$
9:         $\mathcal{P}_{\text{OOD}} = \mathcal{P}_{\text{OOD}} \cup h_\theta(x_{\text{OOD}})$                 ▷ Store $h_\theta$'s ID and OOD predictions
10:     **end for**
11: **end for**
12: **return** $\widehat{\text{Acc}}_{\text{OOD}} = \text{ALine-S/D}\big(\mathcal{P}_{\text{ID}}, \mathcal{Y}_{\text{ID}}, \mathcal{P}_{\text{OOD}}\big)$                 ▷ Apply ALine-S/D

---

CIFAR10 and ImageNet against their Gaussian-Noise corruptions. The resulting TTA performances vary according to the different hyperparameter values, and they seem sensitive particularly in terms of learning rates and batch sizes. Still, these results, improved or degraded, lie on the same positive correlation line in both agreement and accuracy. Such trends are of significant practical value in TTA, where leveraging models with different architectures may necessitate training them separately in advance. Instead, by simply using different values of hyperparameters on a single model, we can eliminate the need to train multiple models.

## 3 RELIABLE TEST-TIME ADAPTATION

Based on the observations, we investigate the enhancement of existing TTA methods in three ways: (i) accurate estimation of OOD accuracy, (ii) calibration after TTA, and (iii) reliable hyperparameter optimization – all performed without assuming access to the labels.

### 3.1 ACCURACY ESTIMATION

In this section, we illustrate how the strong AGL trend shown among TTAed models enables the precise accuracy estimation on target OOD data. We employ the estimation method called ALine-S and ALine-D, presented in Baek et al. (2022), but utilize them within the context of TTA models. Specifically, In each iteration of TTA, after updating model parameters, we inference a batch of OOD and ID data and store their predictions to acquire the predictions for the entire ID and OOD test set. After running such procedures on a series of models, we calculate their accuracy as well as agreement between them using predictions. Then we employ ALine-S/D to estimate their OOD accuracy based on the strong correlations between ID and OOD. The details of our utilization are described in Algorithm 1. This approach yields improved accuracy estimation compared to vanilla models and other estimation baselines (Table 1). Additionally, we investigate the forecastability of TTAed models using the estimated accuracy, across various distribution shifts (Table 2). Details of ALine-S/D are provided in appendix A.2.

**Strong ACL and AGL in TTA lead to precise accuracy estimation.** Table 1 reports the mean absolute error (MAE) between actual and the estimated accuracy for both vanilla and the adapted models after various TTA baselines. We evaluate them on widely used benchmarking distribution shifts in TTA literature, including CIFAR10-C, CIFAR100-C, ImageNet-C (averaged over corruptions), and ImageNet-R. Of particular note is that, especially in CIFAR10-C and CIFAR100-C, applying ALine-S/D on TTAed models achieve substantially lower MAE compared to that of vanilla models (*e.g.*, $5.17\% \rightarrow 0.53\%$ of ALine-D after TENT on CIFAR10-C). This suggests that application of ALine-S/D can be substantially more *effective* in estimating accuracy of the TTAed models, primarily attributed to their *stronger AGL* than vanilla models. We also compare them with other estimation baselines, such as average thresholded confidence (ATC) (Garg et al., 2021), difference of confidence (DOC)-feat (Guillory et al., 2021), average confidence (Hendrycks & Gimpel, 2017), and agreement (Jiang et al., 2022). We find that across different datasets, ALine-S and ALine-D consistently outperform existing baselines on estimating TTA performance.

**TTA effectiveness is forecastable given OOD datasets in the wild.** Here, we extend to various datasets that include common corruptions in CIFAR10-C, CIFAR100-C, ImageNet-C, and real-

| Dataset | Method | ATC | DOC-feat | AC | Agreement | ALine-S | ALine-D |
|---|---|---|---|---|---|---|---|
| | Vanilla | 9.26 | 14.44 | 16.74 | 7.70 | 5.50 | 5.17 |
| | SHOT | 2.91 | 5.32 | 9.94 | 1.48 | 0.73 | **0.56** |
| CIFAR10-C | TENT | 9.21 | 5.32 | 10.08 | 1.51 | 0.73 | **0.53** |
| | ETA | 10.18 | 5.10 | 11.02 | 1.50 | 0.71 | **0.56** |
| | SAR | 1.14 | 4.71 | 7.76 | 1.79 | 0.89 | **0.77** |
| | Vanilla | 5.20 | 10.42 | 14.35 | 15.07 | 8.37 | 8.20 |
| | SHOT | 4.49 | 6.57 | 18.34 | 4.87 | 0.84 | **0.69** |
| CIFAR100-C | TENT | 19.17 | 6.53 | 21.96 | 5.16 | 0.82 | **0.67** |
| | ETA | 25.40 | 4.10 | 28.32 | 6.18 | **0.70** | 1.07 |
| | SAR | 1.85 | 3.48 | 16.68 | 4.54 | 0.90 | **0.77** |
| | Vanilla | 4.27 | 13.47 | 17.76 | 22.28 | 5.87 | 5.87 |
| | SHOT | 5.83 | 4.46 | 9.84 | 14.74 | 4.17 | **4.13** |
| ImageNet-C | TENT | 10.92 | 6.30 | 20.25 | 13.74 | **4.16** | 4.18 |
| | ETA | 6.34 | 5.94 | 23.56 | 13.20 | **4.00** | 4.21 |
| | SAR | 7.04 | **3.86** | 11.00 | 12.27 | 4.31 | 5.25 |
| | Vanilla | 1.72 | 14.31 | 17.74 | 21.56 | 7.62 | 7.62 |
| | SHOT | 5.29 | 12.91 | 17.85 | 17.92 | 3.22 | **3.22** |
| ImageNet-R | TENT | 9.14 | 14.94 | 24.10 | 18.30 | 3.44 | **3.44** |
| | ETA | 11.71 | 12.61 | 34.12 | 17.49 | 2.33 | **2.17** |
| | SAR | 3.78 | 9.44 | 13.49 | 16.11 | 2.82 | **2.44** |

Table 1: Mean absolute error (MAE) (%) of the accuracy estimation on TTAed models with different architectures. ALine-S/D on TTAed models leads to substantially lower errors in accuracy estimation, compared to ALine-S/D applied on the vanilla models as well as other estimation baselines.

| Dataset | SHOT | | TENT | | ETA | | SAR | |
|---|---|---|---|---|---|---|---|---|
| | GT | Est. | GT | Est. | GT | Est. | GT | Est. |
| CIFAR10C-Snow | +3.94 | +4.32 | +4.09 | +4.45 | +4.51 | +4.69 | +2.54 | +2.80 |
| CIFAR100C-Bright | +6.59 | +8.28 | +6.79 | +8.24 | 7.55 | +8.72 | +7.34 | +8.29 |
| ImageNetC-Gauss | +24.53 | +13.45 | +25.79 | +14.64 | +30.25 | +22.82 | +31.28 | +22.19 |
| ImageNet-R | +6.94 | +2.70 | +6.19 | +2.52 | +10.35 | +5.30 | +8.9 | +2.44 |
| CIFAR10.1 | -2.30 | -1.51 | -2.10 | -1.68 | -2.25 | -1.33 | -2.30 | -1.56 |
| CIFAR10.2 | -1.70 | -1.64 | -1.80 | -1.65 | -1.40 | -0.66 | -1.90 | -1.44 |
| ImageNetV2 | -0.27 | -2.50 | -1.18 | -3.42 | -10.34 | -12.60 | -0.21 | -2.40 |
| FMoW-WILDS | -0.22 | -0.73 | +0.37 | +1.03 | +0.29 | +0.70 | +0.62 | +0.85 |

Table 2: Actual (GT) and estimated (Est.) improvement/degradation (%) in OOD accuracy after applying each TTA method w.r.t their base counterparts. The values with green indicate the improvement, whiled red is the degradation. Our estimations consistently have the same predictions (*i.e.*, colors) on whether TTA methods enhance or diminish accuracy across distribution shifts, enabling to forecast their generalizations without labeled data.

world shifts, such as ImageNet-R, CIFAR10.1, CIFAR10.2, ImageNetV2, and FMoW-WILDS. In Table 2, we present actual and estimated OOD accuracy improvement or degradation by each TTA baseline, including SHOT, TENT, ETA, and SAR, w.r.t. their base counterparts. Specifically, we first obtain the estimated accuracy of both TTAed models and vanilla models, and calculate the differences between them.

Our results, in columns of "GT", show that TTA methods sometimes fail to enhance generalization under real-world shifts, e.g. CIFAR10.1, CIFAR10.2 and ImageNetV2. Notably, by observing the trend of accuracy drops (red) in estimated OOD accuracy from the "Est." columns, we can readily identify such failures, without access to any labels. Moreover, the overall trends of generalization on FMoW-WILDS by TTA methods precisely align with those revealed in the estimated results, *i.e.*, all TTA baselines except SHOT improves accuracy. This underscores the potential of our accuracy estimation method for TTA in Algorithm 1, which offers practical guidance for selecting or determining the suitability of TTA methods across diverse shifts, without access to any labels.

## 3.2 UNSUPERVISED CALIBRATION

**Proposed method.** In this section, we introduce a variant of temperature-scaling (Guo et al., 2017) that calibrates models only *with the estimated accuracy* of the model, not the labeled data. Specifically, let $X$ be the random variable for data and $f(X)$ be the logit output of the neural network

| Method | CIFAR10-C | | | CIFAR100-C | | | ImageNet-C | | |
|---|---|---|---|---|---|---|---|---|---|
| | Uncalib. | Ours | Oracle | Uncalib. | Ours | Oracle | Uncalib. | Ours | Oracle |
| Vanilla | 17.48 | 9.71 | 3.42 | 15.25 | 15.82 | 3.09 | 14.97 | 7.92 | 1.79 |
| BN_Adapt | 7.99 | 2.73 | 2.49 | 9.59 | 2.85 | 2.37 | 3.21 | 1.98 | 1.67 |
| TENT | 7.76 | 3.11 | 2.77 | 13.40 | 2.10 | 2.06 | 7.38 | 4.56 | 2.94 |
| ETA | 7.72 | 3.13 | 2.74 | 15.66 | 4.93 | 4.48 | 12.80 | 8.23 | 7.40 |
| Vanilla* | 21.80 | 12.99 | 3.96 | 25.70 | 15.66 | 2.90 | 30.32 | 11.63 | 2.03 |
| ConjPL | 11.89 | 4.43 | 3.56 | 25.48 | 3.01 | 3.00 | 17.13 | 5.50 | 3.84 |

Table 3: Expected calibration error (ECE) (%) of the vanilla, BN_Adapt, TENT, ETA, and ConjPL without calibration method (named as "Uncalib."), using our method (named as "Ours") and the oracle lower-bound using ground-truth labels (named as "Oracle"). Vanilla and Vanilla* denote pretrained models with cross-entropy loss and polyloss respectively. After various TTA methods, our unsupervised calibration method significantly reduces the ECE compared to that of vanilla, manifesting negligible gap comparable to oracle-bound results.

$f$ given input X for classifying among $c$ categories. We define a simple root-finding problem that finds an optimal temperature value $\tau$ that scales the model's averaged confidence to match to the estimated accuracy $\text{Acc}_{\text{est}}$. This can be written as

$$\text{Find } \tau \text{ s.t. } \mathbb{E}\left[\max_c \text{softmax}\left(\frac{f(X)}{\tau}\right)\right] = \text{Acc}_{\text{est}} \tag{2}$$

We use Newton's method solve the root finding problem in Equation 2. Once the optimal $\tau$ is found, we can then temperature scale the prediction using this value.

Note that, instead of the negative log-likelihood loss used in the original temperature scaling method (Guo et al., 2017), our approach minimizes discrepancies between the model's averaged confidence and accuracy over the entire test set. This formulation bears analogy to the definition of the Expected Calibration Error (ECE) metric (Guo et al., 2017), which calculates (a weighted average of) similar discrepancies but now within multiple confidence intervals. Consequently, our method can be regarded as directly minimizing calibration error by utilizing estimated accuracy across the entire bin. Furthermore, it's worth noting that we can use our proposed temperature scaling method during test-time without labelled data, while the original temperature scaling method relies on a held-out validation set.

**Experimental results.** In Table 3, we present a comparison of the calibration error between vanilla and adapted models using various TTA methods, including BN_Adapt, TENT, ETA, and ConjPL. Note that for ConjPL, we use the pretrained model with polyloss (Leng et al., 2022), thus we add "Vanilla" and "Vanilla*" that indicate the vanilla models pretrained with cross-entropy and polyloss, respectively. For each model, we provide ECE values for (i) no calibration, (ii) calibration using our method, and (iii) calibration using an oracle approach. The "oracle" approach represents the lower-bound of the best achievable ECE through temperature scaling, where we find $\tau$ by sweeping over the grids of temperature candidates using ground-truth labels. In this experiment, we employ different checkpoints of a source-trained model, one of the hyperparameters we observed for having strong ACL and AGL (Figure 3).

The results demonstrate that adaptations via minimizing entropy can result in worse calibration, as evidenced by the "Uncalib." results for CIFAR100-C and ImageNet-C. In these cases, TENT, ETA, and ConjPL lead to an increase in ECE compared to that of BN_Adapt. This miscalibration indicates that models make overconfident predictions, particularly on samples that are incorrectly classified. This is mainly attributed to entropy minimization employed in TENT, ETA, and ConjPL, applied across all samples regardless of their correctness (Prabhu et al., 2021; Chen et al., 2022). For TTAed models, our method effectively addresses such issue, and substantially reduces their calibration error to levels close to the lower-bound achieved by the oracle in every dataset. Interestingly, when we apply our method to vanilla models, where the presence of ACL and AGL is less pronounced, a substantial disparity still remains in the calibration results between our method and the lower bound represented by the oracle. This emphasizes the effectiveness of our unsupervised calibration method specifically in the setting of TTA where we observe prominent ACL and AGL.

| HyperParameter | TTA Method | | | | | | |
|---|---|---|---|---|---|---|---|
| | BN_Adapt | SHOT | TENT | ETA | ConjPL | SAR | TTT |
| Learning Rate | – | 0.65 | 0.72 | 0.72 | 0.42 | 0.24 | 3.71 |
| Adapt Step | – | 0.23 | 0.23 | 0.24 | 0.12 | 0.43 | 0.04 |
| Architecture | 0.21 | 0.03 | 0.03 | 0.01 | 0.04 | 0.20 | 0.49 |
| Batch Size | 0.0 | 0.73 | 0.77 | 0.77 | 0.18 | 0.06 | – |
| Checkpoints | 0.0 | 0.07 | 0.05 | 0.01 | 0.11 | 0.01 | 0.48 |

Table 4: Mean Absolute Error (MAE) (%) between accuracy of models selected by our method vs. ground-truth, tested on the CIFAR10-C dataset. Since BN_Adapt do not involve parameter updates, we exclude learning rate and the adaptation steps. In addition, TTT uses a single batch, and we exclude the batch size as well. Our analysis demonstrates that across TTA methods, our hyperparameter selection method consistently identifies hyperparameters that lead to TTAed models achieving OOD accuracy levels comparable to those selected using ground-truth labels.

### 3.3 RELIABLE HYPERPARAMETER OPTIMIZATION

Given the ACL phenomena of TTAed models across varying hyperparameter values, as depicted in Figure 3, *selecting the best-performing model on ID data* emerges as a straightforward and effective strategy for model selection in OOD shifts. While this strategy has been acknowledged in a few prior studies (Miller et al., 2021; Wenzel et al., 2022), our contribution lies in its novel application for optimizing hyperparameters within the TTA framework. Notably, hyperparameter tuning has remained a significant challenge even in the most recent TTA studies (Boudiaf et al., 2022; Zhao et al., 2023), owing to the absence of OOD test data. To find the best hyperparameters, we first explore a wide range of hyperparameter candidates, by systematically sweeping through various values for each parameter. Then, we simply select the models that best performs in ID.

Table 4 reports the OOD accuracy gap between the models selected by our approach vs. best-performing in OOD using ground-truth labels, tested on CIFAR10-C. The results show that across different corruptions as well as TTA baselines, selecting the best-performing hyperparameters in ID consistently results in negligibly high OOD accuracy, with less than $1\%$ MAE compared to those selected with labeled data. Even for sensitive hyperparameters, such as learning rates and batch sizes, where incorrect selections can lead to significant performance degradation (as shown in Figure 3), our approach consistently identifies near-optimal parameters for different TTA baselines. Still, in TTT where we vary different learning rates, our method fails to select optimal hyperparameters, resulting in large MAE. This failure is primarily attributed to the absence of a strong ACL among models using varying learning rates, which is evidenced by Figure 4. We will discuss this failure case further in Section 5.

## 4 RELATED WORK

**Test-time adaptation and its pitfalls of reliability** Test-time adaptation (TTA) enhances model robustness by adapting models to unlabeled test data. One research direction uses self-supervision tasks during both training and testing (Sun et al., 2020; Liu et al., 2021; Gandelsaman et al., 2022), while another explores "fully" test-time adaptations that require no specific pretraining procedures, relying on objectives such as entropy minimization (Wang et al., 2021; Niu et al., 2022; 2023), data augmentation invariance (Zhang et al., 2022; Chen et al., 2022), and self-training with pseudo-labels (Rusak et al., 2022; Goyal et al., 2022; Wang & Wibisono, 2023).

Some studies have extended their evaluation beyond corruptions to include more challenging shifts, such as datasets reproductions (Liu et al., 2021; Zhang et al., 2022), domain generalization benchmarks (Iwasawa & Matsuo, 2021; Zhao et al., 2023), and WILDS (Rusak et al., 2022). Yet, as pointed out in Zhao et al. (2023), TTA methods may not effectively address the full spectrum of distribution shifts in the wild. Another persistent issue is that TTA, especially those based on entropy minimization, can lead to overconfident predictions. Specifically, while Nado et al. (2021) showed that using test-time batch statistics enhances calibration under distribution shifts (Ovadia et al., 2019), several work (Rusak et al., 2022; Chen et al., 2022) observed that entropy minimization diminishes this effect. Moreover, the performance of TTA methods can be severely impacted by the negligent selection of hyperparameters such as learning rates (Boudiaf et al., 2022; Zhao et al., 2023), adaptation steps (Zhao et al., 2023), or batch sizes (Niu et al., 2023; Khurana et al., 2022). A recent study (Boudiaf et al., 2022) addresses this issue by adapting only the model's outputs in-

stead of its parameters. Our study leverages the remarkable observation of the *agreement-on-the-line phenomena within TTA*, offering promising solutions to these reliability issues.

**Accuracy and agreement-on-the-line** Miller et al. (2021) empirically demonstrated that there exists a strong correlation between models' ID and OOD accuracy, namely accuracy-on-the-line (ACL), on various types of distribution-shifts benchmarks, encompassing the variants of CIFAR10, ImageNet and WILDS Sagawa et al. (2022). Such correlations persist across models with varying architectures, hyperparameters, and other setups. Recently, analogous strong correlations based on agreement between models, agreement-on-the-line (AGL), were observed by Baek et al. (2022). Most importantly, they found that when AGL holds, ACL also holds. In addition, their linear lines have almost the same slope and bias when tested across diverse range of datasets, which lead to precise accuracy estimation methods without access to the OOD labels.

However, the underlying reason for such phenomenon remain unclear, and certain datasets reveal weak accuracy and agreement-on-the-line trends, as observed in CIFAR10 test vs. Gaussian Noise corruption in CIFAR10-C. Recent work (Wenzel et al., 2022; Teney et al., 2023; Liang et al., 2023) have investigated the broader types of distribution shifts, identifying shifts that show different trends beyond the linear correlations. In this study, we investigate the impact of model adaptation on reinforcing (or maintaining) linear correlations, and identify novel conditions that lead to such trends–adapting with varying TTA hyperparameters.

## 5 LIMITATIONS

Even though we consistently observe the linear trends across various models and distribution shifts during TTA, we also find an exceptional case where such trends do not manifest. As shown in Figure 4 right, we empirically observe that varying learning rates in TTT (Sun et al., 2020) exhibits that the ID and OOD accuracies are negatively correlated, leading to a complete misalignment with the agreement line. While in other experimental setups, such as using different architectures, we still observe the strong AGL and ACL among the models, as shown in Figure 4 left. Such negative correlations results

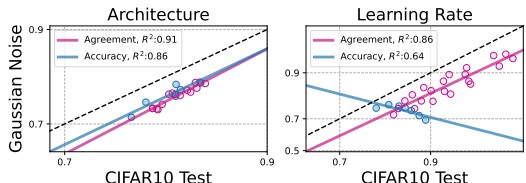

Figure 4: Compared to the adapted models with varying architectures, using learning rates in TTT (Sun et al., 2020) shows negative correlations in accuracies, resulting in misalignment between agreement and accuracy lines.

in the accuracy of the models selected using our method, *i.e.*, best-performing ID accuracy, suffer significant deviation from the best-performing OOD accuracy, as shown in Table 4.

## 6 CONCLUSION AND DISCUSSIONS

In this work, we observe that adapting models during test time maintains or even reinforces the strong linear ID vs. OOD correlations in their accuracy and agreement, across models and distributions. Such observations motivate us to enhance the TTA methods' reliability in surprisingly simple but effective ways, specifically from three perspectives: accurate performance estimation of TTA methods, calibration of confidence in TTAed models, and reliable hyperparameter tuning, all without need for labels. We conduct extensive experiments across distribution shifts, TTA baselines and their hyperparameters, and demonstrate the effective estimation of the TTAed models' performances. We then forecast their successes (as well as their possible failures) of generalization under various shifts including real-world datasets. Moreover, we perform model calibration and reliable hyperparameter optimization, achieving results comparable to the level of those assuming access to the ground-truth labels.

Our results show that existing adaptation strategies under distribution shifts lead to strong AGL and ACL linear trends. This naturally raises questions about how to theoretically characterize the conditions under which adaptations enhance these linear trends. We believe this is a promising future research directions that can ascertain the reliable observations of AGL, leading to reliable TTA across *any* types of distribution shifts. Additionally, we recognize that, to observe AGL and ACL, access to ID test data (and its labels for ACL) is required. This might raise privacy concerns due to the potential inclusion of sensitive information, and also demand additional computation resources. Therefore, overcoming such dependencies on ID data and exploring "fully" unsupervised approach of observing AGL and ACL remain promising directions for future research, e.g., leveraging foundation models.

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

# A APPENDIX

## A.1 EXPERIMENTAL SETUP

This section provides the details regarding distribution shifts, network architectures, TTA baselines and their hyperparameters we used in the paper.

**Distribution shifts.** We test the models on $8$ different distribution shifts that include synthetic corruptions and real-world shifts. Synthetic corruptions datasets, CIFAR10-C, CIFAR100-C, and ImageNet-C (Hendrycks & Dietterich, 2019) are designed to contain the $15$ different types of corruptions, such as Gaussian noise, on their original dataset counterparts. We use the most severe corruptions, severity of $5$, in all experiments in our paper. These corruptions datasets are most commonly evalutated distribution shifts in a wide range of TTA papers (Schneider et al., 2020; Sun et al., 2020; Liu et al., 2021; Liang et al., 2020; Wang et al., 2021; Goyal et al., 2022; Niu et al., 2022; 2023; Zhao et al., 2023).

We also test on real-world shifts, which include CIFAR10.1 (Recht et al., 2018), CIFAR10.2 (Lu et al., 2020), ImageNetV2 (Recht et al., 2019), ImageNet-R (Hendrycks et al., 2021), and FMoW-WILDS (Sagawa et al., 2022). CIFAR10.1, CIFAR10.2, and ImageNetV2 are the reproduced datasets of their base counterparts by following the original dataset creation procedures. ImageNet-R is the variant of ImageNet and it contains the images with renditions of various styles, such as paintings or cartoons. FMoW (Sagawa et al., 2022) contains the spatiotemporal satellite imagery of $62$ different use of land or building categories, where distribution shifts originate from the years that the imagery is taken. Specifically, following Miller et al. (2021), we use ID set consists of images taken from 2002 to 2013, and OOD set taken between 2013 and 2016.

**Network architectures.** We use a different set of the network architectures for specific datasets and their shifts. Specifically, for CIFAR10 and CIFAR100 vs. their OOD shifts, we use ResNet-18,26,34,50,101 (He et al., 2016), WideResNet-28-10 (Zagoruyko & Komodakis, 2017), Mo-bileNetV2 (Sandler et al., 2018), VGG11,13,16,19 (Simonyan & Zisserman, 2015). For ImageNet and FMoW-WILDS vs. their OOD shifts, we leverage ResNet-18,34,50,101,152, WideResNet-50,101, DenseNet121 (Huang et al., 2017), ResNeXt50-32x4d (Xie et al., 2016). As mentioned in Section 2.1, we use the pretrained model weights from torchvision, except for TTT (Sun et al., 2020) and ConjPL (Goyal et al., 2022). In addition, since TTT requires the rotation-prediction task during pretraining on source data, we train them ourselves using ResNet14,26,32,50,104, and 152, which are available in their original implementation[2] and use them for Section 3.3.

**Optimizer and learning rates.** We use SGD optimizer with momentum of $0.9$ for all TTA baselines except for SAR, which uses sharpness-aware minimization (SAM) optimizer (Foret et al., 2021). For Table 1, we evaluate and observe the linear trends among the models with different architecture, adapted with the same learning rates. Then we average their results across different learning rates. Also in Figure 3 and Table 4, we evaluate the models adapted with different learning rates, and observe the linear trends. For these experiments, we sweep over in the grid of $\{10^{-4}, 2 \cdot 10^{-4}, 5 \cdot 10^{-4}, 10^{-3}, 2 \cdot 10^{-3}, 5 \cdot 10^{-3}, 10^{-2}\}$ for CIFAR10, CIFAR100, and their OOD shifts, while using $\{5 \cdot 10^{-5}, 10^{-4}, 2 \cdot 10^{-4}, 5 \cdot 10^{-4}, 10^{-3}, 2 \cdot 10^{-3}, 5 \cdot 10^{-3}\}$ for ImageNet and their OOD counterparts. For Figures 1, 2 and Table 2, we use the fixed learning rates of $10^{-3}$ for CIFAR10, CIFAR100 and their OOD counterparts, and $2.5 \cdot 10^{-4}$ for ImageNet, FMoW and their counterparts.

**Batch sizes.** In Figure 3 and Table 4, we use test a wide spectrum of different values of batch size, which we sweep over the grid of $\{1, 2, 4, 8, 16, 32, 64, 128, 256, 512\}$. For the other experiments, we fix the batch size of $128$ for TTA on CIFAR10, CIFAR100 and their OOD, and $64$ on ImageNet, FMoW for their OOD. Note that for TTT, we use a single batch following the original paper setup.

**Number of adapt steps.** In Table 3.3, we sweep through one to five adaptation steps for every TTA baseline we use. For the rest of the experiments, we use single step of the adaptations. Also, we utilize the online adaptations (*i.e.*, no initialization for each batch) for all baselines.

---

[2]https://github.com/yueatsprograms/ttt_cifar_release

---

**Algorithm 2** ALine-S and ALine-D

---

1: **Input:** ID predictions $\mathcal{P}_{\text{ID}}$ and labels $\mathcal{Y}_{\text{ID}}$, OOD predictions $\mathcal{P}_{\text{OOD}}$.
2: **Function:** Probit transform $\Phi^{-1}(\cdot)$, Linear regression $\mathcal{F}(\cdot)$.
3: ─────────────────────────────────────
4: $\hat{a}, \hat{b} = \mathcal{F}(\Phi^{-1}(\text{Agr}(\mathcal{P}_{\text{ID}})), \Phi^{-1}(\text{Agr}(\mathcal{P}_{\text{OOD}})))$         ▷ Estimate slope and bias of linear fit
5: $\widehat{\text{Acc}}_{\text{OOD}}^{\text{S}} = \Phi(\hat{a} \cdot \text{Acc}(\mathcal{P}_{\text{ID}}, y_{\text{ID}}) + \hat{b})$                      ▷ ALine-S
6: Initialize $\boldsymbol{A} \in \mathbb{R}^{\frac{n(n-1)}{2} \times n}, \boldsymbol{b} = \mathbb{R}^{\frac{n(n-1)}{2}}$
7: i=0
8: **for** $(p_{j,\text{ID}}, p_{k,\text{ID}}), (p_{j,\text{OOD}}, p_{k,\text{OOD}}) \in \mathcal{P}_{\text{ID}}, \mathcal{P}_{\text{OOD}}$ **do**
9:     $\boldsymbol{A}_{ij} = \frac{1}{2}, \boldsymbol{A}_{ik} = \frac{1}{2}, \boldsymbol{A}_{il} = 0 \forall l \notin j, k$
10:     $\boldsymbol{b}_i = \Phi^{-1}(\text{Agr}(p_{j,\text{OOD}}, p_{k,\text{OOD}})) + \hat{a} \cdot \big( \frac{\Phi^{-1}(\text{Acc}(p_{j,\text{ID}}, y_{\text{ID}})) + \Phi^{-1}(\text{Acc}(p_{k,\text{ID}}, y_{\text{ID}}))}{2} - \Phi^{-1}(\text{Agr}(p_{j,\text{ID}}, p_{k,\text{ID}})) \big)$
11:     i=i+1
12: **end for**
13: $\boldsymbol{w}^* = \arg\min_{\boldsymbol{w} \in \mathbb{R}^n} ||\boldsymbol{Aw} - \boldsymbol{b}||_2^2$
14: $\widehat{\text{Acc}}_{\text{OOD}}^{\text{D}} = \Phi(\boldsymbol{w}_i^*) \forall i \in [n]$                               ▷ ALine-D
15: **return** $\widehat{\text{Acc}}_{\text{OOD}}^{\text{S}}, \widehat{\text{Acc}}_{\text{OOD}}^{\text{D}}$

---

## A.2   OOD ACCURACY ESTIMATION METHODS

**ALine-S and ALine-D**   Baek et al. (2022) propose ALine-S and ALine-D, which assess the models' OOD accuracy without access to labels by leveraging the agreement-on-the-line among models. We provide the detailed algorithm of ALine-S and ALine-D in Algorithm 2.

**Average thresholded confidence (ATC)**   Garg et al. (2021) introduce OOD accuracy estimation method, ATC, which learns the confidence threshold and predicts the OOD accuracy by using the fraction of unlabeled OOD samples for which model's negative entropy is less that threshold. Specifically, let $h(x) \in \mathbb{R}^c$ denote the softmax output of model $h$ given data $x$ from $\mathcal{X}_{\text{OOD}}$ for classifying among $c$ classes. The method can be written as below:

$$\widehat{\text{Acc}}_{\text{OOD}} = \mathbb{E}\left[ \mathbb{1}\{s(h(x)) < t\} \right], \tag{3}$$

where $s$ is the negative entropy, *i.e.*, $s(h(x)) = \sum_c h_c(x) \log(h_c(x))$, and $t$ satisfies

$$\mathbb{E}\left[ \mathbb{1}\{s(h(x)) < t\} \right] = \mathbb{E}\left[ \mathbb{1}\{ \arg\max_c h_c(x) \neq y \} \right]. \tag{4}$$

**Difference of Confidence (DOC)-feat**   Guillory et al. (2021) observe that the shift of distributions is encoded in the difference of model's confidences between them. Based on this observation, they leverage such differences in confidences as the accuracy gap under distribution shifts for calculating the final OOD accuracy. Specifically,

$$\widehat{\text{Acc}}_{\text{OOD}} = \text{Acc}_{\text{ID}} - \left( \mathbb{E}\left[ \max_c h_c(x_{\text{ID}}) \right] - \mathbb{E}\left[ \max_c h_c(x_{\text{OOD}}) \right] \right) \tag{5}$$

**Average Confidence (AC)**   Hendrycks & Gimpel (2017) estimate the OOD accuracy based on model's averaged confidence, which can be written as

$$\widehat{\text{Acc}}_{\text{OOD}} = \mathbb{E}\left[ \max_c h(x_{\text{OOD}}) \right]. \tag{6}$$

**Agreement**   Jiang et al. (2022) observe that disagreement between the models that are trained with different setups closely tracks the error of models in ID. We adopt this as the baseline for assessing generalization under distribution shifts, where we can estimate $\widehat{\text{Acc}}_{\text{OOD}} = \text{Agr}(\mathcal{P}_{\text{OOD}})$, where $\mathcal{P}_{\text{OOD}}$ denotes the set of predictions of the models on OOD data $\mathcal{X}_{\text{OOD}}$.

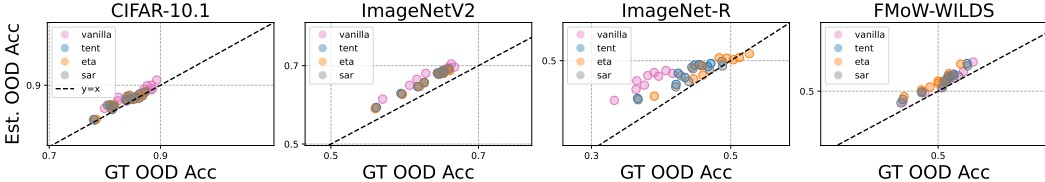

Figure 5: Comparison of GT vs. estimated OOD accuracy of vanilla and TTAed models under real-world shifts. Each pink dot represents vanilla, blue the TENT, orange the ETA, and gray the SAR results. The dotted $y = x$ line denotes the *perfect* estimation line, where the closer dots are located to the line, the more accurate the estimations are.

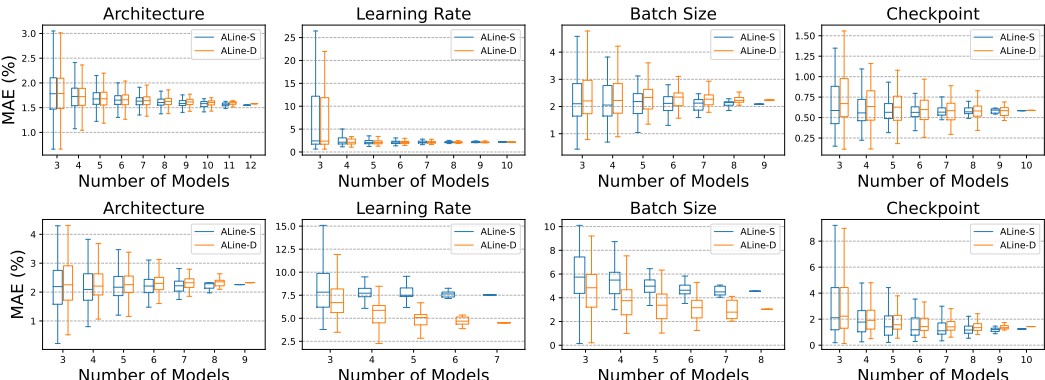

Figure 6: Distribution of MAE of accuracy estimation over different number of models. We test models adapted with different TTA setups. The results on first row are those adapted by TENT on CIFAR10-C Gaussian Noise, and those on second row are adapted by ETA on ImageNet-C Gaussian Noise.

### A.3 ACCURACY ESTIMATION RESULTS ON REAL-WORLD SHIFTS.

In addition to the strong AGL and ACL trends that persist under real-world shifts (Figure 2), we demonstrate their accurate estimated OOD accuracy by comparing them with the ground-truth accuracy on such shifts. In Figure 5, we test the models adapted with TENT, ETA, and SAR, and show that their estimated results are close to the GT OOD accuracy (*i.e.*, closer to $y = x$) across datasets. In particular, in ImageNetV2 and ImageNet-R, the results of TTAed models are located closer to $y = x$ than those of vanilla, indicating TTAed models' accurate estimation performances more than those of vanilla models.

### A.4 ANALYSIS ON THE NUMBER OF TTAED MODELS FOR ACCURATE ESTIMATION.

In this section, we investigate how the accuracy estimation performances of ALine-S/D change as we vary the number of the models used in estimation. To this end, we examine the same models used in searching for hyperparameters in Section 3.3, and here we test different architectures, learning rates, batch sizes, and the checkpoints of the source-trained models. We vary the size of models, $n$, and for each size, we calculate all possible sets of models' estimation error.

Figure 6 illustrates the distributions of the estimation error in MAE, w.r.t. the number of models used. We test TENT on CIFAR10 test vs. CIFAR10-C Gaussian Noise, and ETA on ImageNet test vs. ImageNet-C Gaussian Noise. We observe that with minimum number of models for ALine-S/D, which is three, the estimation results have low values of MAE, especially in using different architectures or the checkpoints of the source-trained models. In addition, we note that further improvements in estimation accuracy, evidenced by a decrease in MAE, are easily attainable by adding only a small number of additional models. This effect is particularly pronounced when considering learning rates for both datasets and batch size for ImageNet-C. These results indicate that during TTA, accurate estimation of OOD accuracy can be achieved with a reasonable number

| HyperParameter | Ratio of ID data (%) | | | | | | |
|---|---|---|---|---|---|---|---|
| | 1 | 2 | 5 | 10 | 20 | 50 | 100 |
| Architecture | $1.46\pm0.38$ | $1.51\pm0.46$ | $1.56\pm0.42$ | $1.53\pm0.46$ | $1.65\pm0.40$ | $1.63\pm0.20$ | 0.49 |
| Learning Rate | $3.27\pm1.06$ | $2.80\pm1.09$ | $2.23\pm0.75$ | $1.93\pm0.59$ | $2.04\pm0.40$ | $1.98\pm0.22$ | 1.95 |
| Adapt Step | $2.79\pm1.16$ | $2.20\pm0.82$ | $1.73\pm0.60$ | $1.68\pm0.47$ | $1.55\pm0.35$ | $1.52\pm0.18$ | 1.54 |
| Batch Size | $3.14\pm0.97$ | $2.63\pm0.87$ | $2.26\pm0.65$ | $2.28\pm0.47$ | $2.27\pm0.33$ | $2.22\pm0.18$ | 2.23 |
| Checkpoints | $2.76\pm0.70$ | $2.50\pm0.63$ | $1.91\pm0.43$ | $1.53\pm0.43$ | $1.13\pm0.29$ | $0.75\pm0.15$ | 0.59 |

Table 5: Mean Absolute Error (MAE) (%) of accuracy estimation by our method, tested on the CIFAR10 vs. CIFAR10-C Gaussian Noise. Each column includes the results of our method using only the portion (%) of the ID data, ranging from 1% to 100% (full access). The 100 random subsets of ID samples are tested, and corresponding mean and standard deviation of the results are reported. Results show that even with a critically limited ID data, our method robustly estimates TTA accuracy comparable to those with full access.

| Dataset | Ratio of ID data (%) | | | | | | | |
|---|---|---|---|---|---|---|---|---|
| | Before | 1 | 2 | 5 | 10 | 20 | 50 | 100 |
| CIFAR10-C | 7.76 | $3.79\pm0.45$ | $3.83\pm0.62$ | $3.65\pm0.30$ | $3.57\pm0.23$ | $3.57\pm0.18$ | $3.64\pm0.39$ | 3.11 |
| ImageNet-C | 7.38 | $4.7\pm1.64$ | $4.02\pm0.97$ | $4.47\pm1.02$ | $4.36\pm0.48$ | $4.63\pm0.39$ | $4.68\pm0.26$ | 4.56 |

Table 6: Expected Calibration Error (ECE) (%) of the calibration by our method, tested on CIFAR10-C and ImageNet-C (averaged over 15 corruptions). Each column includes the results of our method using only the portion (%) of the ID data, ranging from 1% to 100% (full access). The 100 random subsets of ID samples are tested, and corresponding mean and standard deviation of the results are reported. Results show that even with a critically limited ID data, our method robustly calibrates TTA overconfident predictions, comparable to those with full access.

of models, each with different hyperparameter values, thereby offering practical feasibility during testing.

## A.5 ANALYSIS ON THE NUMBER OF ID SAMPLES.

In real-world scenarios, access to the ID data can be critically limited in TTA setup, which might raise the privacy concerns and require excessive computing resources. In this section, we investigate the feasibility and effectiveness of our methods under conditions where the access to ID samples is critically limited. Specifically, when applying our method to TENT on CIFAR10-C and ETA on ImageNet-C Gaussian Noise, we vary the ratio of ID samples from 100% (full) to 1%. We then iterate over multiple runs (i.e., 100 for CIFAR10 and 10 for ImageNet) with randomly selected subsets of ID and report the mean / standard deviation of the accuracy estimation and the calibration results.

As shown in Tables 5 and 6, our methods demonstrate the robust performances in accuracy estimation and calibration, showing negligible performances gap from those assuming the full access to ID data. For instance, in Table 5, results of using just 1% of total ID data across various setups achieves comparably low MAE against the ones using 100% of ID samples. Moreover, in Table 6, our methods with only 1% of ID samples successfully calibrates the ETA-adapted models, demonstrating decreased ECE close to those with full access.

## A.6 ADDITIONAL EXPERIMENTAL RESULTS

**Strong ACL and AGL after TTA.** In this section, we supplement the Section 2 by providing additional results of TTAed models exhibiting strong AGL and ACL linear trends across datasets and TTA methods. Specifically, in Figures 7 and 8, we provide the results of SHOT, TENT, ETA, and SAR comparing with that of vanilla, across every corruption types of CIFAR10-C with highest severity. The axes are probit-scaled, and each blue and pink dot represent the accuracy and agreement of the model, respectively. In Figure 9, we add the results on CIFAR10.2 along with other TTA methods' results on the real-world shifts. We consistently observe that AGL and ACL consistently

persist or even become stronger than vanilla models, when applying various TTA methods across distribution shifts, evidenced by the high $R^2$ values in all cases.

**Forecasting TTA effectiveness given OOD datasets in the wild.** In Table 7, we supplement the results of Table 2 by providing the relative improvements/degradation of accuracy after TTA across every corruption in CIFAR10-C, CIFAR100-C, and ImageNet-C. Our results show that the the effectiveness of various TTA baselines can be predicted via our accuracy estimation methods on TTAed models across diverse corruption types.

**Accuracy estimation using adaptation hyperparameters.** To supplement the results of Table 1, we apply our methods in accuracy estimation by utilizing various adaptation hyperparameter values, which include learning rates, adapt steps, batch sizes, and checkpoints. As shown in Table 8, our methods consistently show superior accuracy estimation results across setups, datasets, when compared to the estimation baselines.

| Dataset | | SHOT | | TENT | | ETA | | SAR | |
|---|---|---|---|---|---|---|---|---|---|
| | | GT | Est. | GT | Est. | GT | Est. | GT | Est. |
| CIFAR10C | Gaussian | +39.17 | +32.04 | +39.55 | +32.11 | +40.02 | +32.78 | +37.53 | +29.74 |
| | Shot | +36.87 | +29.13 | +36.87 | +29.23 | +37.53 | +29.43 | +34.81 | +26.92 |
| | Impulse | +46.38 | +24.68 | +46.33 | +24.51 | +47.18 | +24.63 | +44.84 | +22.43 |
| | Defocus | +1.14 | +0.95 | +1.21 | +1.03 | +1.3 | +0.93 | +0.82 | +0.31 |
| | Glass | +15.68 | +14.61 | +15.71 | +14.61 | +15.53 | +14.61 | +14.01 | +13.68 |
| | Motion | +6.18 | +4.9 | +6.28 | +4.93 | +6.39 | +5.12 | +5.19 | +4.12 |
| | Zoom | +0.97 | +0.36 | +1.11 | +0.47 | +0.98 | +0.36 | +0.71 | +0.17 |
| | Snow | +3.94 | +4.32 | +4.09 | +4.45 | +4.51 | +4.69 | +2.54 | +2.80 |
| | Frost | +11.35 | +10.81 | +11.4 | +10.88 | +11.18 | +10.87 | +10.2 | +9.65 |
| | Fog | +19.63 | +11.24 | +19.67 | +11.18 | +20.25 | +12.04 | +15.33 | +7.29 |
| | Brightness | +0.88 | +1.61 | +0.85 | +1.43 | +0.86 | +1.44 | +0.3 | +1.02 |
| | Contrast | +34.41 | +22.7 | +34.44 | +22.85 | +34.6 | +22.95 | +32.91 | +20.53 |
| | Elastic | +2.37 | +2.53 | +2.29 | +2.81 | +2.61 | +3.04 | +1.27 | +1.53 |
| | Pixelate | +10.05 | +8.76 | +10.07 | +8.75 | +10.23 | +9 | +9.49 | +8.28 |
| | Jpeg | +6.32 | +3.44 | +6.36 | +3.68 | +6.88 | +3.82 | +3.61 | +0.77 |
| CIFAR100C | Gaussian | +42.35 | +16.06 | +41.94 | +16.32 | 42.39 | +15.9 | +42.24 | +16.03 |
| | Shot | +40.95 | +16.36 | +40.13 | +15.78 | 40.2 | +15.01 | +40.96 | +15.09 |
| | Impulse | +41.93 | +22.66 | +41.14 | +22.6 | +40.35 | +22.09 | +41.12 | +20.8 |
| | Defocus | +5.56 | +3.19 | +5.16 | +2.82 | +5.7 | +3.85 | +6.4 | +3.94 |
| | Glass | +30.05 | +15.91 | +29.49 | +15.2 | +29.47 | +15.35 | +29.2 | +15.66 |
| | Motion | +13.35 | +10.26 | +13.32 | +9.68 | +13.64 | +10.19 | +13.84 | +10.1 |
| | Zoom | +4.71 | +2.87 | +4.38 | +2.45 | +4.42 | +2.41 | +4.72 | +3.26 |
| | Snow | +15.08 | +12.48 | +14.57 | +11.79 | 14.29 | +11.58 | +15.32 | +12.72 |
| | Frost | +26.57 | +16.98 | +26.25 | +16.21 | +26.58 | +16.99 | +26.77 | +18.13 |
| | Fog | +33.32 | +20.73 | +32.69 | +19.9 | +32.38 | +19.91 | +32.98 | +20.04 |
| | Bright | +6.59 | +8.28 | +6.79 | +8.24 | 7.55 | +8.72 | +7.34 | +8.29 |
| | Contrast | +54.92 | +27.61 | +54.92 | +27.85 | +55.07 | +28.16 | +54.68 | +27.65 |
| | Elastic | +10.88 | +8.3 | +10.49 | +8.06 | +10.36 | +8.19 | +11.51 | +8.69 |
| | Pixelate | +24.47 | +19.2 | +23.96 | +18.52 | +24.13 | +18.31 | +24.54 | +19.17 |
| | Jpeg | +13.58 | +10.68 | +13.23 | +9.85 | +12.88 | +9.61 | +13.94 | +10.78 |
| ImageNetC | Gauss | +24.53 | +13.45 | +25.79 | +14.64 | +30.25 | +22.82 | +31.28 | +22.19 |
| | Shot | +25.44 | +19.77 | +29.71 | +20.2 | +29.15 | +20.33 | +29.63 | +21.21 |
| | Impulse | +25.46 | +17.6 | +30.22 | +17.45 | +32.79 | +21.36 | +31.65 | +19.66 |
| | Defocus | +7.95 | +3.87 | +11.21 | +3.09 | +7.01 | +2.37 | +12.45 | +3.84 |
| | Glass | +15.28 | +1.31 | +18.46 | +0.86 | +18.15 | +2.8 | +20.7 | +0.79 |
| | Motion | +22.85 | +16.77 | +29.15 | +16.94 | +32.53 | +19.74 | +29.86 | +17.86 |
| | Zoom | +24.71 | +15.34 | +28.17 | +16.34 | +29.92 | +17.94 | +28.61 | +16.5 |
| | Snow | +27.34 | +16.64 | +31.9 | +16.84 | +34.23 | +19.36 | +32.39 | +17.66 |
| | Frost | +16.67 | +11.27 | +18.52 | +10.06 | +22.23 | +13.32 | +20.52 | +11.99 |
| | Fog | +31.28 | +19.74 | +34.08 | +20.85 | +35.35 | +22.14 | +34.35 | +20.88 |
| | Brightness | +7.83 | +6.88 | +8.75 | +7.14 | +8.95 | +7.54 | +8.77 | +7.2 |
| | Contrast | +23.83 | +9.39 | +17.14 | +2.34 | +20.58 | +5.45 | +30.74 | +8.42 |
| | Elastic | +35.53 | +17.57 | +39.3 | +18.35 | +40.71 | +19.25 | +39.58 | +18.4 |
| | Pixelate | +35.75 | +36.95 | +38.85 | +37.85 | +40.01 | +39.28 | +39.01 | +37.94 |
| | Jpeg | +18.26 | +21.79 | +22.0 | +23.22 | +23.2 | +24.78 | +22.2 | +23.17 |
| ImageNet-R | | +6.94 | +2.70 | +6.19 | +2.52 | +10.35 | +5.30 | +8.9 | +2.44 |
| CIFAR10.1 | | -2.30 | -1.51 | -2.10 | -1.68 | -2.25 | -1.33 | -2.30 | -1.56 |
| CIFAR10.2 | | -1.70 | -1.64 | -1.80 | -1.65 | -1.40 | -0.66 | -1.90 | -1.44 |
| ImageNetV2 | | -0.27 | -2.50 | -1.18 | -3.42 | -10.34 | -12.60 | -0.21 | -2.40 |
| FMoW-WILDS | | -0.22 | -0.73 | +0.37 | +1.03 | +0.29 | +0.70 | +0.62 | +0.85 |

Table 7: Actual (GT) and estimated (Est.) improvement/degradation (%) in OOD accuracy after applying each TTA method w.r.t their base counterparts, including every corruption of CIFAR10-C, CIFAR100-C, ImageNet-C, as well as real-world shifts, such as ImageNet-R, CIFAR10.1, CIFAR10.2, ImageNetV2, and FMoW-WILDS. The values with green indicate the improvement, whiled red is the degradation. Our estimations consistently have the same predictions (*i.e.*, colors) on whether TTA methods enhance or diminish accuracy across distribution shifts, enabling to forecast their generalizations without labeled data.

| Dataset | Method | ATC | DOC-feat | AC | Agreement | ALine-S | ALine-D |
|---|---|---|---|---|---|---|---|
| CIFAR10-C | Learning rate | 7.53 | 5.13 | 9.49 | 8.60 | 2.25 | **2.22** |
| | Adapt Step | 5.14 | 5.77 | 10.17 | 8.65 | **1.26** | 1.33 |
| | Batch Size | 2.82 | 4.36 | 41.14 | 5.16 | **1.31** | 1.34 |
| | Checkpoint | 1.60 | 4.52 | 7.16 | 3.70 | **0.42** | 0.43 |
| CIFAR100-C | Learning rate | 12.14 | 5.66 | 18.34 | 16.86 | 3.60 | **3.52** |
| | Adapt Step | 14.16 | 11.48 | 27.33 | 18.15 | 2.01 | **2.01** |
| | Batch Size | 2.44 | 4.62 | 51.05 | 3.73 | 1.37 | **1.25** |
| | Checkpoint | 1.31 | 4.90 | 11.22 | 12.09 | **0.72** | 0.83 |

Table 8: Mean absolute error (MAE) (%) of the accuracy estimation on TTAed models with different hyperparameters, including learning rate, adapt step, batch size, and checkpoint. ALine-S/D on TTAed models leads to substantially lower errors in accuracy estimation when compared to existing accuracy estimation baselines.

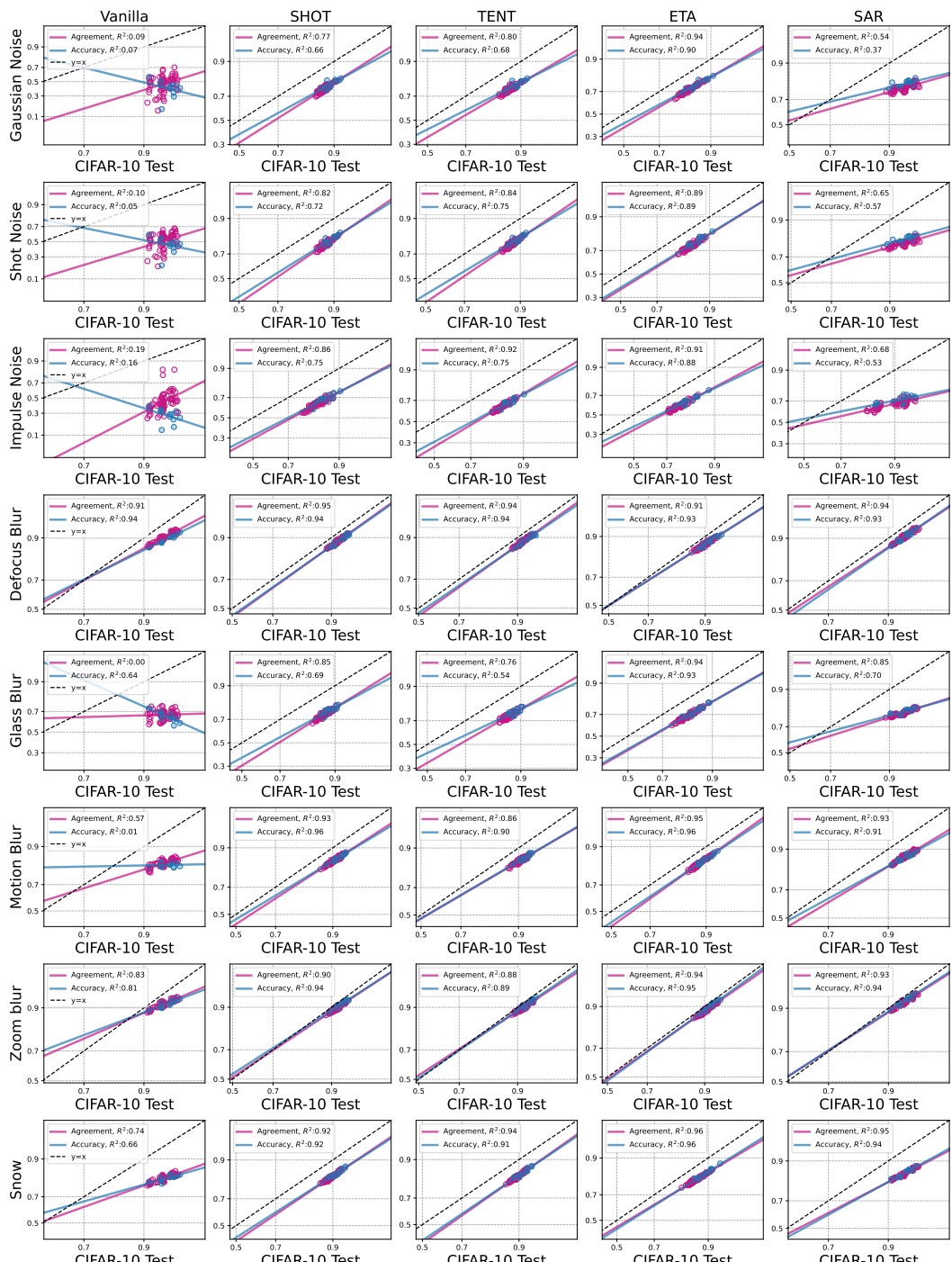

Figure 7: Additional results across every corruption types of CIFAR10-C, where models after TTA show strong ACL and AGL than vanilla models.

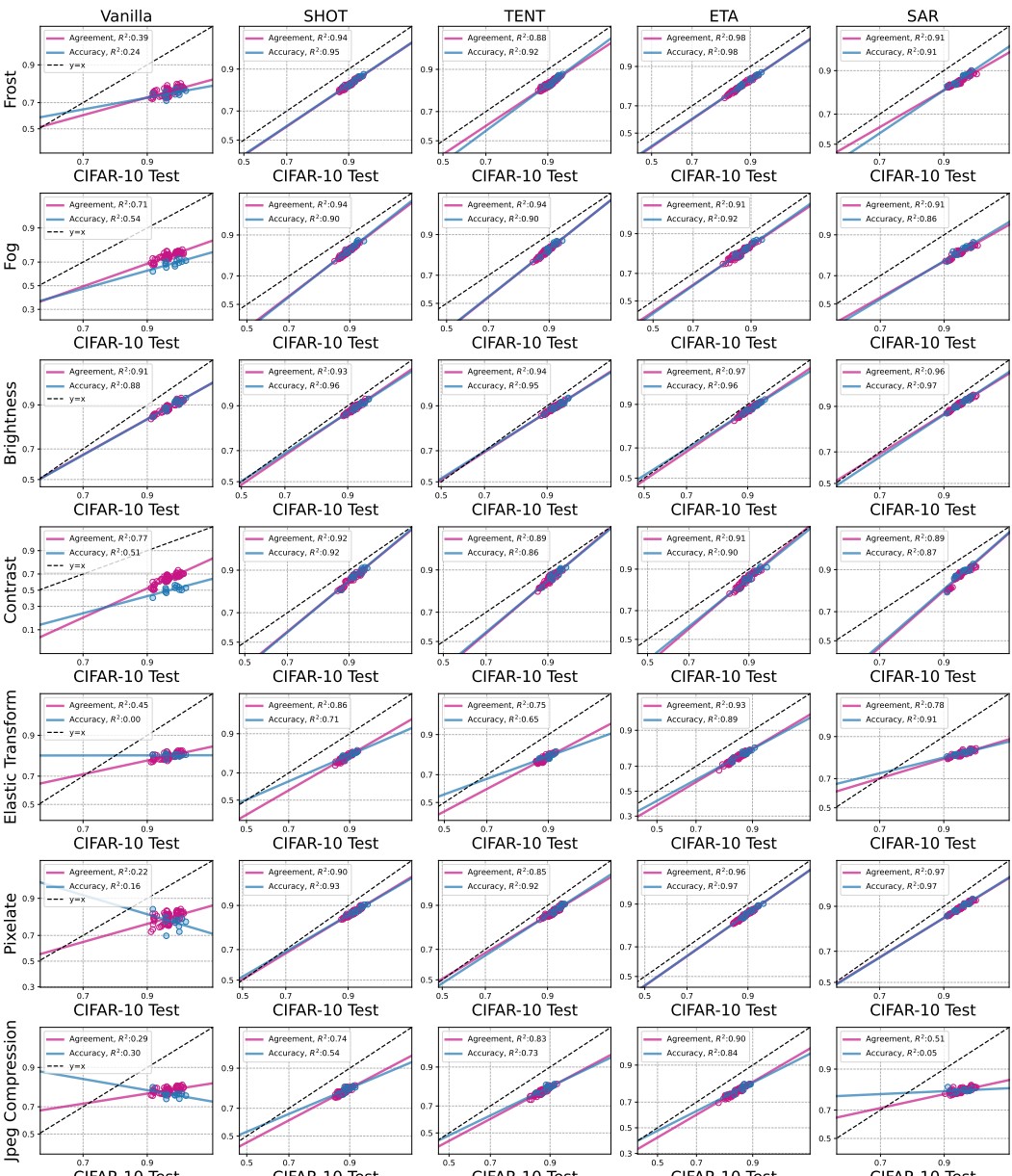

Figure 8: Additional results across every corruption types of CIFAR10-C, where models after TTA show strong ACL and AGL than vanilla models.

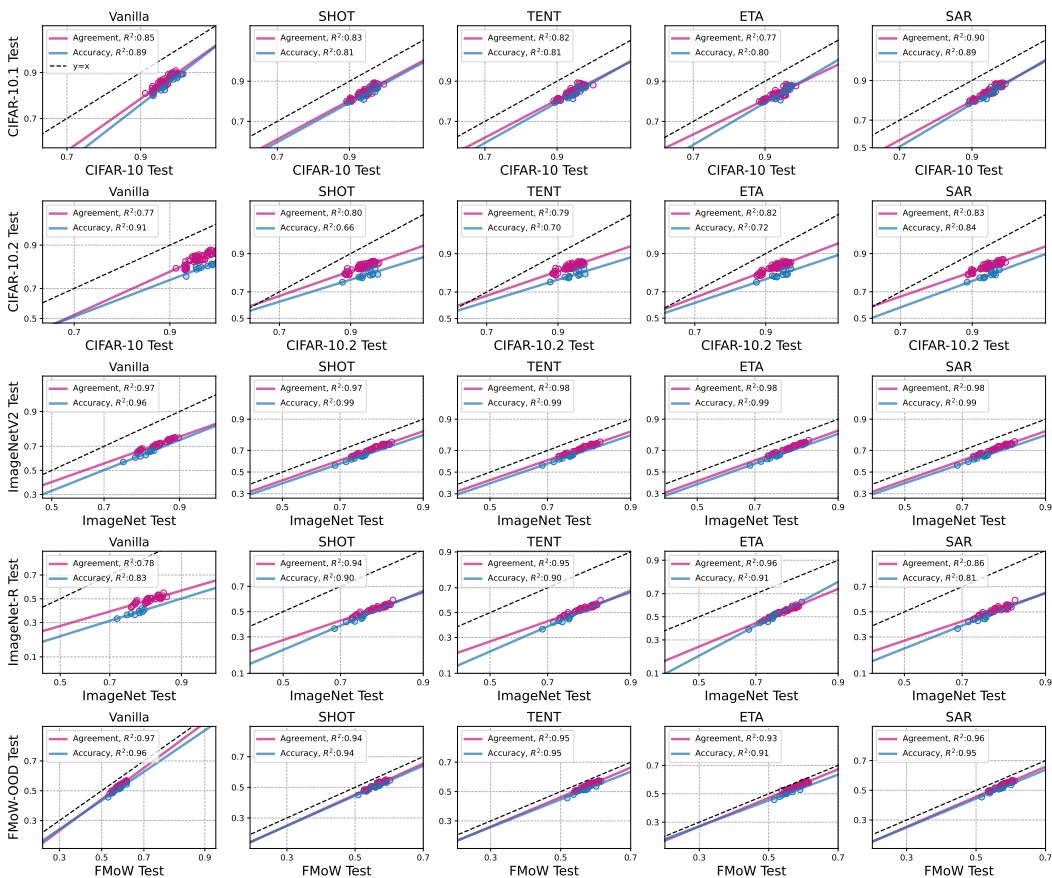

Figure 9: Additional TTA baselines' results under real-world shifts, CIFAR10.1, CIFAR10.2, ImageNetV2, ImageNet-R, and FMoW-WILDS, where models after TTA persist strong ACL and AGL.

