# OpenReview forum: "Reliable Test-Time Adaptation via Agreement-on-the-Line"
_ICLR.cc/2024/Conference — Submitted to ICLR 2024_

### Official Review · Reviewer_gmmt · 2023-10-27

**Soundness:** 2 fair
**Presentation:** 3 good
**Contribution:** 3 good
**Rating:** 5
**Confidence:** 4

**Summary:**

This paper investigates a practical and valuable issue of Test-Time Adaptation, that is, how to pre-estimate the TTA performance of TTAed models. The authors found that a strong correlation existed between TTAed models’ OOD performance/agreement and ID performance/agreement. Based on these findings, the authors further proposed methods for the TTAed model’s calibration and hyperparameter selection. The overall findings are novel but still not convincing enough, necessitating further justifications. So I currently give a borderline score and I will re-evaluate the paper after rebuttal. My detailed comments are as follows.

**Strengths:**

The studied problem is invaluable in TTA but often overlooked by the current community. As many TTA methods modify model parameters during inference and may suffer from an instability issue, in real-world applications, it is essential to pre-estimate the Adaptation Performance to determine whether TTA is needed or how to achieve the best TTA performance using only unlabeled test data.

The observed correlation between TTA performance and ID performance is novel and interesting. The resulting methods for Calibration and Hyper-parameter Selection are simple yet effective.

**Weaknesses:**

1. The claim of “OOD and ID performance/agreement correlation” requires more empirical evidence. Please refer to Question 1.
2. The proposed methods rely on full access to the ID data and work in an offline manner, which slightly weakens its application scenarios.
3. Algorithm 1 requires to maintenance of $n$ models for TTA, which may be memory-consuming and inefficient.

**Questions:**

1. Could the authors include more results regarding more advanced network models in Figures 1 -3, including but not limited to VisionTransformers(Tiny/Small/Base/Large and with different input resolutions) and  SwinTransformers from the Timm repository? These model architectures are quite different from the considered ones and are also trained with more advanced strategies (such as data augmentation, stochastic depth, EMA, dropout, etc.) I am wondering whether the “correlation” still holds under these scenarios.

2. Are there any (or potential) solutions that can modify the proposed method to be an online version? In other words, how about the performance of Algorithm 1 work in an online setting?

3. Are there any sensitivity analyses about $n$ in Algorithm 1?

4. For Calibration and Hyper-parameter Selection, how many ID samples are needed? Will this ID sample number affect the performance significantly?

5. How about the performance of the proposed methods under wild test settings proposed by SAR [ICLR 2023]？

6. If possible, I am also curious about the ID-OOD correlation of MEMO [Memo: Test time robustness via adaptation and augmentation]  and Contrastive Learning Objectives [Contrastive Test-Time Adaptation].

---

> ### Author Response · Authors · 2023-11-20
> **Response to Reviewer gmmt [1/2]**
>
> >### Could the authors include more results regarding more advanced network models in Figures 1 -3, including but not limited to VisionTransformers(Tiny/Small/Base/Large and with different input resolutions) and SwinTransformers from the Timm repository?
>
> We hope our responses in [here](https://openreview.net/forum?id=iEFMwP5wng&noteId=vZZSNTMlWr) clarified the reviewer's concern about the empirical evidence of the OOD and ID performance/agreement correlation.
>
> ---
>
> >### Are there any (or potential) solutions that can modify the proposed method to be an online version? In other words, how about the performance of Algorithm 1 work in an online setting?
>
> We hope our responses in [here](https://openreview.net/forum?id=iEFMwP5wng&noteId=pAkj6nKxb2) sufficiently addressed the concern about the offline nature of the proposed methods.
>
> ---
>
> >### Are there any sensitivity analyses about $n$ in Algorithm 1?
>
> We hope our responses in [here](https://openreview.net/forum?id=iEFMwP5wng&noteId=pAkj6nKxb2) sufficiently addressed the concern about the sensitivity of our method to the number of models $n$.
>
> ---
>
> >### For Calibration and Hyper-parameter Selection, how many ID samples are needed? Will this ID sample number affect the performance significantly?
>
> We examined the sensitivity of our method on the number of ID samples [here](https://openreview.net/forum?id=iEFMwP5wng&noteId=3aU3nNw8Bt), and we hope this sufficiently addressed the concern.

---

> > ### Author Response · Authors · 2023-11-22
> > **Response to Reviewer gmmt [2/2]**
> >
> > >### How about the performance of the proposed methods under wild test settings proposed by SAR [ICLR 2023]？
> >
> > Following the wild test settings suggested by Niu et al. (2023) [1], we examined the persistence of AGL/ACL under three setups: 1) mixed shifts, 2) single-batch, and 3) label-shift. As suggested by Niu et al. (2023), we utilized transformer-based backbone models with layer normalization (LN) layers, and we applied SAR on ImageNet-C. Specifically, we followed the default hyperparameter, such as learning rates as in Niu et al. (2023), used the highest imbalance ratio (i.e., 500000), and tested
> > * ViT [2]: ViT-B/16, ViT-L/16, ViT-S/32, ViT-B/32
> > * Swin Transformer [3]: Swin-S, Swin-B, Swin-L
> >
> > **Even in these wild setups, we consistently observed strong AGL/ACL among models with different architectures** (https://tinyurl.com/3zs7vat6), with high $R^2$ values, when tested in ImageNet-C Gaussian Noise. These results demonstrate that the AGL/ACL phenomena is broadly observed across various settings, including the unconventional ones proposed by Niu et al. (2023).
> >
> > Additionally, we assessed the accuracy estimation of these models using our method, averaged over architectures and every corruption in ImageNet-C, as shown in below. The resulting MAE (%) of ALine-D is 5.27 in mixed shift, and 5.72 in label-shift, which show precise estimation results comparable to those tested in the normal setup.
> >
> > | Method | Mixed Shift | Label Shift |
> > | ---------- | ---------- | ---------- |
> > | ALine-S | 6.15 | 7.59 |
> > | ALine-D | 5.27 | 5.72 |
> >
> > Since adapting in single-batch setup is much slower than other setups, we will add the results during the remaining rebuttal period or in the camera-ready version.
> > We will add the results as well as the discussions under the wild setup in our camera-ready version.
> >
> > [1] Niu et al., “Towards Stable Test-Time Adaptation in Dynamic Wild World”, ICLR’23
> >
> > [2] Dosovitskiy et al., “An image is worth 16x16 words: transformers for image recognition at scale”, ICLR’21
> >
> > [3] Liu et al., “Swin Transformer: Hierarchical Vision Transformer using Shifted Window”, ICCV’21
> >
> > >### If possible, I am also curious about the ID-OOD correlation of MEMO [Memo: Test time robustness via adaptation and augmentation] and Contrastive Learning Objectives [Contrastive Test-Time Adaptation]
> >
> > We tested MEMO [1] with the same set of architectures in Figure 1 on CIFAR10-C Gaussian Noise. The results show **strong correlated lines in accuracy and agreement** (https://tinyurl.com/4tbnrsv2), confirming the consistency of AGL/ACL phenomena across different TTA baselines. The MAE (%) of accuracy estimation on models adapted with MEMO on CIFAR10-C Gaussian Noise is 1.16 % for ALine-S and 1.16 % for ALine-D.
> >
> > AdaContrast [2] employs contrastive learning with online pseudo-labeling for target domains. Due to the limited time in the rebuttal, we could not implement the baseline to see if AGL/ACL persist in the method. We will make sure to include AdaContrast as one of our TTA baselines to observe the linear trends in our camera-ready version.
> >
> > [1] Zhang et al., "MEMO: Test Time Robustness via Adaptation and Augmentation", NeurIPS'22
> >
> > [2] Chen et al., "Contrastive Test-Time Adaptation", CVPR'22

---

> > ### Comment · Reviewer_gmmt · 2023-11-23
> > **Further questions**
> >
> > Thanks for the authors's extensive response. Most of my concerns have been addressed but I am still unsure how you modify the method to an online version. Could you please provide more details and give a pseudo-code/algorithm?

---

> ### Author Response · Authors · 2023-11-23
> **Response to Reviewer gmmt - Details on Online Version Method**
>
> We appreciate the reviewer's feedback, and we are pleased to provide more details about the modifications made in our method for an online setup. The algorithms (https://tinyurl.com/5n72hev6) directly compare the online version (**above**) and the offline-version (**below**). Blue-colored comments highlight the newly added detail from the algorithm in the original submission.
>
> A key modification is that **we apply ALine-S/D only to the predictions of current ID and OOD batch at each iteration (Line 10 in Algorithm 1) in the online version**. In this case, $\mathcal P_\text{ID}$ and $\mathcal P_\text{OOD}$ contain predictions of $n$ models on the current test batch ($x_\text{ID}$ and $x_\text{OOD}$), and ALine-S/D is applied to these.
>
> On the contrary, in the offline version, we apply ALine-S/D at the end of the algorithm (Line 11 in Algorithm 2) after saving predictions on the entire test set ($\mathcal X_\text{ID}$ and $\mathcal X_\text{OOD}$) in $\mathcal P_\text{ID}$ and $\mathcal P_\text{OOD}$, with ALine-S/D then applied to these aggregated predictions.
>
> Please let us know if there are any additional questions.

---

### Official Review · Reviewer_72ab · 2023-10-27

**Soundness:** 3 good
**Presentation:** 3 good
**Contribution:** 3 good
**Rating:** 6
**Confidence:** 4

**Summary:**

This study addresses critical challenges in Test-Time Adaptation (TTA) methods, used to bolster model robustness against distribution shifts, by leveraging unlabeled data from the test distribution. The authors identify a consistent "agreement-on-the-line" phenomenon in adapted models, regardless of varied hyperparameters and across diverse distribution shifts. Utilizing this insight, they enhance TTA reliability by introducing strategies for estimating Out-Of-Distribution (OOD) accuracy, recalibrating models, and tuning hyperparameters without labeled test data. Comprehensive experiments validate these approaches, demonstrating evaluation of TTA methods and improvements in both OOD accuracy and calibration error, akin to scenarios having access to ground-truth labels.

**Strengths:**

Highlighting a Critical Problem: The researchers focused on the test-time adaptation (TTA) issue, showing that current methods aren't great at evaluating how well TTA methods work. This work is important because it tells us we need better ways to check if these methods are doing what we want them to do without always having to rely on labeled data.

Interesting Observation about Model Behavior: They found something pretty unexpected in the models after TTA. These models tended to show a "line agreement" behavior more than they did before. It means that after trying to adapt the models to new situations, they started to behave in a predictable pattern, which was interesting and useful to know for future work.

**Weaknesses:**

**Insufficient Evidence for OOD and ID Performance Correlation Claims**: The authors' assertions about model behavior correlations lack broad empirical support. The research predominantly revolves around CNNs, which is just one structure among the diverse architectures and applications in deep learning. To substantiate these findings' universality and efficacy, it's crucial to extend the studies to other network models and datasets, such as Vision Transformers (ViT) and Swin Transformers, and more complex tasks like object detection and semantic segmentation. Confirming similar observations across a wider range of contexts would instill greater confidence in the authenticity and applicability of this phenomenon.

**Heavy Reliance on Full Access to In-Distribution Data**: The methods proposed in the study assume complete access to in-distribution training data, an assumption impractical in many real-world scenarios. Constraints like computational resources or data privacy issues might restrict access to comprehensive in-distribution training data, leaving practitioners with only the trained model. Under these circumstances, the proposed evaluation method may become inapplicable, losing its value. The authors need to tackle this limitation, potentially necessitating the design of novel experiments or methods that support evaluation and adaptation in these more restrictive environments.

**Discrepancy Between the Offline Nature of Methods and Practical Needs**: The methods introduced in this research operate within a static, offline setting, overlooking the dynamic, continuous arrival of data in real-world applications. In real-world scenarios, models must accommodate online or incremental data streams, known as online test-time adaptation. The researchers should contemplate this online setting and investigate whether their strategies remain effective when handling continuous data flows. This exploration might require new experimental setups and adaptability tests to simulate the reception conditions of data in the real world.

**Questions:**

Please address the concerns above.

---

> ### Author Response · Authors · 2023-11-20
> **Response to Reviewer 72ab**
>
> >### Insufficient Evidence for OOD and ID Performance Correlation Claims (Transformers, More complex tasks like object detection and semantic segmentation)
>
> We additionally included Vision Transformers (ViT) and Swin Transformers as our models to observe AGL/ACL in [here](https://openreview.net/forum?id=iEFMwP5wng&noteId=vZZSNTMlWr).
>
> To the best of our understanding, the utilization of agreement to assess the generalization of models is well-established in the existing literature [1,2,3,4,5], particularly in the context of image classification tasks. Our work aligns with this convention, concentrating on leveraging the AGL phenomena to enhance reliability of TTA in image classification. However, we agree with the reviewer’s point that it is promising to extend the exploration of these phenomena to more complex tasks beyond image classification. We will make sure to augment our discussions on this aspect in the camera-ready version of our work.
>
> [1] Madani et al., “Co-Validation: Using Model Disagreement on Unlabeled Data to Validate Classification Algorithms”, NeurIPS’04
>
> [2] Nakkiran and Bansal, “Distributional generalization: A new kind of generalization”, Arxiv’21
>
> [3] Jiang et al., “Assessing generalization of SGD via disagreement”, ICLR’22
>
> [4] Baek et al., Agreement-on-the-Line: Predicting the Performance of Neural Networks under Distribution Shift, NeurIPS’22
>
> [5] Lee et al., “Demystifying Disagreement-on-the-Line in High Dimensions”, ICML’23
>
> ---
>
> >### Heavy Reliance on Full Access to In-Distribution Data
>
> We hope our responses in [here](https://openreview.net/forum?id=iEFMwP5wng&noteId=3aU3nNw8Bt) clarified the reviewer's concern about the reliance on full access to ID data.
>
> ---
>
> >### Discrepancy Between the Offline Nature of Methods and Practical Needs
>
> We hope our responses in [here](https://openreview.net/forum?id=iEFMwP5wng&noteId=pAkj6nKxb2) sufficiently addressed the concern about the offline nature of the proposed methods.

---

### Official Review · Reviewer_a5t4 · 2023-10-31

**Soundness:** 2 fair
**Presentation:** 3 good
**Contribution:** 2 fair
**Rating:** 5
**Confidence:** 4

**Summary:**

The authors aim to achieve reliable test-time adaptation (TTA) by tackling three bottlenecks, including performance evaluation without labeled data, miscalibration after TTA, and hyperparameter selection for TTA methods. Specifically, the authors empirically verify the strong correlation in agreement and accuracy between in-distribution (ID) data and out-of-distribution (OOD) data. Based on this phenomenon, the authors adopt the ALine-S/D method to estimate OOD accuracy based on ID data and labels. With the estimated OOD accuracy, they further calculate the optimal scaling temperature to calibrate the model prediction for a lower expected calibration error. Moreover, the authors select the hyperparameters that enable the adapted model to obtain the best performance on the ID data as the optimal TTA hyperparameters on the OOD data. However, some significant issues are required to be further addressed. My detailed comments are as follows.

**Strengths:**

1. The authors reveal and verify the agreement-on-the-line and accuracy-on-the-line phenomenon in various TTA methods and various datasets, suggesting a new idea of hyperparameter tuning for TTA methods.
2. The authors propose a new evaluation method to estimate accuracy on out-of-distribution (OOD) data without OOD labels, which further enables a more precise model calibration via temperature scaling.
3. Experimental results demonstrate the effectiveness of the proposed unsupervised calibration methods. For example, the proposed unsupervised calibration method is able to reduce the expected calibration error from 13.40 to 2.10 while using TENT under CIFAR100-C.

**Weaknesses:**

1. The authors have empirically observed the occurrence of agreement-on-the-line (AGL) and accuracy-on-the-line (ACL) following test-time adaptation (TTA). To enhance the manuscript, it would be beneficial for the authors to provide a more extensive explanation and discussion regarding this observed phenomenon.
2. The proposed temperature scaling method may not be applicable in latency-sensitive real-world applications when considering efficiency. As described in section 3.2, the optimal temperature is calculated after the network makes predictions on the full test set. Therefore, the proposed method require a significant delay for model adaptation. More discussion on efficiency is required.
3. Although the authors conduct experiments on different TTA methods, the problem setting of this paper is different from that of TTA. As shown in Algorithm 1, the proposed performance estimation method requires in-domain data and labels, which, however, are inaccessible under the settings of TTA.
4. Figures 1-3 are difficult to understand. For example, it is unclear what the values of the horizontal and the vertical axes in Figure 1 represent. More explanations should be provided.
5. The authors only analyze the agreement-on-the-line phenomenon on CNN-based models. However, powerful transformer-based models should also be involved in the experiments, such as ViT[1], Swin Transformer[2], PoolFormer[3], and so on.
6. In the Introduction, the authors claim that "Baek et al. (2022) show that the ID and OOD agreement between classifiers shows a strong linear correlation". However, Baek et al. (2022) only study the correlation of disagreement and error between classifiers. It would be preferable to provide a more precise description.
7. In Table 2, the authors only provide insufficient results on several domains of the test datasets to demonstrate the effectiveness of the proposed performance estimation. It would be better if the authors could provide more experimental results under these datasets, such as all 15 corrupted datasets in ImageNet-C.
8. In Figure 3, the authors demonstrate that the phenomenon of AGL and ACL also occurs when varying TTA hyperparameters, such as learning rate and batch size. However, from Table 1 to Table 3, the authors only study the effect of the proposed methods using different architectures or different training checkpoints. More ablation study using different hyperparameter setup is suggested.
9. As shown in Figure 4, varying learning rate in TTT exhibits a negative correlation between ID and OOD accuracies. To establish a comprehensive study, more experiments should be conducted to verify if this phenomenon occurs in other test-time training methods, such as TTT++[4] and TTT-MAE[5].
10. On page 7, the sentence “let X the random variable” should be changed to “let X be the random variable”.


[1] An Image is Worth 16x16 Words: Transformers for Image Recognition at Scale, ICLR 2021.

[2] Swin Transformer: Hierarchical Vision Transformer using Shifted Windows, ICCV 2021.

[3] PoolFormer: MetaFormer Is Actually What You Need for Vision, CVPR 2022.

[4] TTT++: When Does Self-Supervised Test-Time Training Fail or Thrive? NeurIPS 2021.

[5] Test-Time Training with Masked Autoencoders, NeurIPS 2022.

**Questions:**

Please refer to the Weakness

---

> ### Author Response · Authors · 2023-11-20
> **Response to Reviewer a5t4 [1/2]**
>
> >### The authors have empirically observed the occurrence of agreement-on-the-line (AGL) and accuracy-on-the-line (ACL) following test-time adaptation (TTA). To enhance the manuscript, it would be beneficial for the authors to provide a more extensive explanation and discussion regarding this observed phenomenon.
>
> We have added detailed explanations about their definitions as well as characteristics in the related work (Section 4) in our revised manuscript.
>
> ---
>
> >### Figures 1-3 are difficult to understand. For example, it is unclear what the values of the horizontal and the vertical axes in Figure 1 represent. More explanations should be provided.
>
> The figures mainly show the TTAed models’ accuracy (%) tested on ID (x-axis) and OOD (y-axis), represented as dots with blue color. At the same time, these models’ agreement (%) tested on ID and OOD are plotted in the same x-axis and y-axis, denoted as pink dots. In addition, we plot the blue and pink lines that represent the linear fit of each dot, corresponding to the linear correlations between ID and OOD. In Figures 1 and 2, each dot represents the accuracy/agreement of the models with different architectures, while in Figure 3, they denote that of the models adapted with different adaptation hyperparameters. We provided such detailed explanations on each caption of the figures in the revised version of our submission.
>
> ---
>
> >### In the Introduction, the authors claim that "Baek et al. (2022) show that the ID and OOD agreement between classifiers shows a strong linear correlation". However, Baek et al. (2022) only study the correlation of disagreement and error between classifiers. It would be preferable to provide a more precise description.
>
> We would like to clarify that our original claim in the Introduction was that “Baek et al. (2022)[1] show that the OOD agreement between classifiers shows a strong linear correlation with their ID agreement.” To expound on our intention, we quote the specific sentence from Baek et al.(2022)[1] that encapsulates the core observation: “ Whenever accuracy-on-the-line holds, we observe that the OOD agreement between the predictions of any two pairs of neural networks (with potentially different architectures) also observes a strong linear correlation with their ID agreement.” We clarified our claim in the Introduction (Section 1) of our revised submission.
>
> [1] Baek et al., Agreement-on-the-Line: Predicting the Performance of Neural Networks under Distribution Shift, NeurIPS’22
>
> ---
>
> >### In Table 2, the authors only provide insufficient results on several domains of the test datasets to demonstrate the effectiveness of the proposed performance estimation.
>
> We supplemented Table 2 by providing the performance prediction results across every 15 corruption of CIFAR10-C, CIFAR100-C, and ImageNet-C in Section A.6 and Table 7 of the appendix in our revised submission. The results show that our methods consistently predict the actual effectiveness of the tested TTA methods across diverse corruptions and datasets, without relying on the labels.
>
> ---
>
> >### From Table 1 to Table 3, the authors only study the effect of the proposed methods using different architectures or different training checkpoints. More ablation study using different hyperparameter setup is suggested.
>
> To supplement the incomplete results the reviewer pointed out, we provide the results of accuracy estimation and calibration using different hyperparameter setups, which include learning rates, adapt steps, batch sizes, and checkpoints. In the Table below, we compare the estimation results (MAE (%)) of our methods (on ETA) with other baselines on ImageNet-C (averaged over every corruption). The results show that our methods with different hyperparameters achieve state-of-the-art performances across various hyperparameters and datasets (Results on CIFAR10-C/CIFAR100-C (TENT) are provided in Section A.6 and Table 8 of the appendix in our revised submission).
>
> **ImageNet-C [MAE (\%)]**
> | Setup | ATC | DOC-feat | AC | Agreement | ALine-S | ALine-D
> | ---------- | ---------- | ---------- | ---------- | ---------- | ---------- | ---------- |
> | Learning rate | 5.60 | 5.28 | 37.12 | 18.22 | 4.46 | **3.70**|
> | Adapt step | 11.22 | 9.31 | 31.51 | 32.60 | 6.01 | **5.75** |
> | Batch size | 8.19 | 6.70 | 26.18 | 14.29 | 3.45 | **2.34** |
> | Checkpoint | 2.96 | 5.77 | 12.06 | 21.28 | 3.01 | **2.70** |
>
> We also provide the calibration results using different hyperparameters when applying TENT on CIFAR10-C and CIFAR100-C (averaged over every corruption). Similar to Table 3, our methods effectively reduce the calibration error using the adapt steps, learning rates, or batch sizes.
>
> | Setup | CIFAR10-C | CIFAR100-C |
> | ---------- | ---------- | ---------- |
> | Uncalibrated | 7.76 |13.40 |
> | Learning rate | 3.23 | 3.29 |
> | Adapt step | 3.40 | 4.56 |
> | Batch size | 3.50 | 3.67 |

---

> > ### Author Response · Authors · 2023-11-22
> > **Response to Reviewer a5t4 [2/2]**
> >
> > >### more experiments should be conducted to verify if this phenomenon occurs in other test-time training methods, such as TTT++ and TTT-MAE.
> >
> > Addressing the reviewer’s concern, we tested TTT++ [1] on CIFAR10-C Gaussian Noise. Similar to TTT, TTT++ exhibits negative correlations among models adapted with different learning rates, while holding strong positive correlations between models adapted with different architectures (https://tinyurl.com/ycxsbjap). Notably, these negative correlations are more evident among models adapted with **smaller learning rates** than larger ones. This implies that ***adapting TTT and TTT++ with too small learning rates may lead to models not fully adapted to OOD***, maintaining high ID but still low OOD accuracy, resulting in negative correlations in accuracy.
> >
> > As the reviewer pointed out, additional analysis on such phenomena will enhance our understanding of circumstances where AGL/ACL observations can be guaranteed. Even though we could not provide the results in TTT-MAE [2] in the limited timeline of the rebuttal, we will make sure to conduct such additional studies including TTT-MAE on further analyzing such phenomenon in our camera-ready version.
> >
> > [1] Liu et al., TTT++: "When Does Self-Supervised Test-Time Training Fail or Thrive?", NeurIPS’21
> >
> > [2] Gandelsman et al., "Test-Time Training with Masked Autoencoders", NeurIPS’22
> >
> >
> > >### On page 7, the sentence “let X the random variable” should be changed to “let X be the random variable”.
> >
> > We revised the grammatical error in Section 3.2 of our revised version.

---

### Official Review · Reviewer_mLJy · 2023-10-31

**Soundness:** 3 good
**Presentation:** 3 good
**Contribution:** 3 good
**Rating:** 6
**Confidence:** 3

**Summary:**

Test-time adaptation (TTA) improves model performance in the presence of distribution shifts that happen at test-time, however, its reliability is hard to evaluate as it requires access to labels at test time. To this end, this work proposes to use the, introduced in prior work, agreement-on-the-line phenomenon in order to improve TTA. Agreement between two models, $h$ and $h’$ is defined as $\\mathbb{E}_{x \\sim \\mathcal{D}}[\\mathbb{1}\\{h(x) = h’(x)\\}]$ and the main intuition is that, after updating a model with TTA, there is a linear relationship between the agreement on the in-distribution (i.e., the distribution used to train the original model) and the agreement on the out-distribution, i.e., the distribution shift observed at test time. The authors then use this observation in order to estimate performance on out-of-distribution data after TTA (without needing access to test labels), perform hyperparameter optimization and improve calibration after TTA.

**Strengths:**

- Reliability of TTA is an important topic and thus this work is a relevant and timely contribution
- The results are convincing, so they could be useful in general for future research on TTA
- The experiments are extensive and cover various tasks

**Weaknesses:**

- The novelty of this method is relatively limited; it is an application of Baek et al. (2022) to the TTA setup
- This method requires adapting multiple models during test time at the specific distribution shift in order to compute the agreement and use it to improve TTA. This might limit practical applications where training multiple models is expensive and, furthermore, the improvements on TTA seem to be on hindsight as one first needs to train multiple models in order to understand whether TTA will yield improvement. Furthermore, given that TTA usually operates on a stream of OOD data and provides a stream of predictions, it is unclear how predictions will be made at test time; does one use the ID model, (one of) the TTA models or does one wait up until the agreement-on-the-line has been computed before making any predictions on the stream?
- Some of the details about the method are not clear.

**Questions:**

Overall, I believe this work is a nice contribution in the field of TTA, as it addresses important issues in TTA. Having said that, the novelty is relatively small, hence my rating. As for questions to the authors:
- What is considered to be $h(x)$? Is it the most probable class under $h$? If so, why not define agreement in terms of the entire distribution on the output space for models, e.g., some kind of expected divergence between $h(x)$ and $h’(x)$?
- How many models were used to get the agreement / accuracy lines at the figures?
- At algorithm 1, it seems that predictions on the ID and OOD data are done under a model that is continuously updated on OOD data, therefore, the ordering of the batches might have an effect on the agreement lines. Do the results change significantly under different random seeds?

---

> ### Author Response · Authors · 2023-11-20
> **Response to Reviewer mLJy [1/2]**
>
> > ### The novelty of this method is relatively limited; it is an application of Baek et al. (2022) to the TTA setup.
>
> While acknowledging the insights from Baek et al. (2022), we want to emphasize that our work explores the AGL phenomenon in adaptive setups. In contrast, Baek et al. (2022) primarily focused on pretrained models with their weights unchanged during test time. We specifically identify an unexpected condition where linear trends persist or even **hold stronger**, as evidenced by the extensive results across datasets, TTA baselines, and experimental setups.
>
> Furthermore, our proposed methods for accuracy estimation, calibration, and hyperparameter optimization, represent a novel contribution. These methods facilitate the reliable utilization of TTA, an aspect rarely explored or addressed by previous studies due to a lack of access to OOD labels.
>
> ---
>
> > ### Given that TTA usually operates on a stream of OOD data and provides a stream of predictions, it is unclear how predictions will be made at test time.
>
> We elaborate on the detailed procedure for making predictions during TTA in our method. TTA operates as it usually does: it updates model parameters online with a stream of OOD data. At each iteration, after parameter updates, we perform inference on batches of OOD and ID data, storing their predictions. After running these procedures on multiple models, we calculate their accuracy and agreement using predictions. Subsequently, we utilize ALine-S/D to estimate OOD accuracy based on the AGL/ACL. We have provided the detailed explanation on how we obtain models’ predictions in the ID and OOD during TTA in Section 3.1 of the revised submission.
>
> Moreover, as demonstrated [here](https://openreview.net/forum?id=iEFMwP5wng&noteId=pAkj6nKxb2), our method is applicable in the online setup. To make predictions, for each iteration, we update models with streamed OOD batch and perform inference on both OOD and ID batches to obtain predictions. Instead of storing predictions until the entire test set, we calculate accuracy and agreement on the current batch using AGL/ACL and apply our methods for accuracy estimation and calibration.
>
> ---
>
> >### What is considered to be h(x)? Is it the most probable class under h? If so, why not define agreement in terms of the entire distribution on the output space for models, e.g., some kind of expected divergence between h(x) and h'(x)?
>
> As noted by the reviewer, h(x) denotes the most probable class predicted by the model h given data x. We clarified the definition of h(x) in Section 2.1 of the revised version of our submission.
>
> We want to clarify that in our work, we employed the definition of agreement by following prior studies that have established the concept of (dis)agreement between classifiers to assess their generalizations in both in-domain contexts [1,2,3] and OOD scenarios [4]. These studies empirically showcased the superiority of assessing generalizations of the models via (dis)agreement between models trained in different setups. A recent paper [5] attempted to demystify the effectiveness of employing such (dis)agreement between models for predicting OOD performances within a theoretical framework.
>
> [1] Madani et al., “Co-Validation: Using Model Disagreement on Unlabeled Data to Validate Classification Algorithms”, NIPS’04
>
> [2] Nakkiran and Bansal, “Distributional generalization: A new kind of generalization”, Arxiv’21
>
> [3] Jiang et al., “Assessing generalization of SGD via disagreement”, ICLR’22
>
> [4] Baek et al., Agreement-on-the-Line: Predicting the Performance of Neural Networks under Distribution Shift, NeurIPS’22
>
> [5] Lee et al., “Demystifying Disagreement-on-the-Line in High Dimensions”, ICML’23
>
> ---
>
> >### How many models were used to get the agreement / accuracy lines at the figures?
>
> Here’s the detailed setups for the figures:
> * Figures 1,2: a total of 11 and 9 models of different architectures for CIFAR10-C, and ImageNet-C, respectively.
> * Figure 3 first row: a total of 10 models of different learning rates, five of adapt steps, 9 of batch sizes, and 10 of checkpoints in CIFAR10-C
> * Figure 3 second row: a total of 7 models of different learning rates, five of adapt steps, 8 of batch sizes and 10 of checkpoints in ImageNet-C

---

> > ### Author Response · Authors · 2023-11-20
> > **Response to Reviewer mLJy [2/2]**
> >
> > >### The ordering of the batches might have an effect on the agreement lines. Do the results change significantly under different random seeds?
> >
> > To answer this, we iterated over 10 different random seeds to run TENT on different ordering of batches. Specifically, we tested in the identical setup of Figure 1 of our submission, where we applied TENT on 11 models with different architectures, evaluated on CIFAR10-C Gaussian Noise. In table below, we reported the mean and standard deviation of the AGL/ACL across different orderings, $R^2$, slope and bias of the linear line.
> >
> > We noticed that strong AGL/ACL consistently hold across varying data orders, maintaining consistently high $R^2$ as well as (almost) the same slope and bias between accuracy and agreement correlation lines. Thus, our method estimates their accuracies with consistently low MAE (%), which are **1.67±0.05** and **1.70±0.05**, for ALine-S and ALine-D, respectively. This demonstrates that our method’s superior accuracy estimation is not affected by the different order of batches.
> >
> > **CIFAR10-C**
> > | Method | $R^2$ (AGL) | $R^2$ (ACL) | Slope (AGL) | Slope (ACL) | Bias (AGL) | Bias (ACL) |
> > | ---------- | ---------- | ---------- | ---------- | ---------- |---------- | ---------- |
> > | TENT | 0.90±0.3 | 0.86±0.04 | 1.05±0.06| 0.96±0.04 | -0.53±0.07 | -0.40±0.05|

---

> > > ### Comment · Reviewer_mLJy · 2023-11-22
> > > **Response to rebuttal**
> > >
> > > I appreciate the response from the authors which clarified my questions. I will maintain my score for now and will update appropriately after the discussion with the other reviewers.

---

### Author Response · Authors · 2023-11-20
**Response to All Reviewers [1/3]**

We greatly appreciate the valuable feedback provided by the reviewers regarding our submission.

The reviewers have acknowledged the significance and interest of our paper, particularly in
* Addressing a significant question of the reliability of recent test-time adaptation (TTA), often overlooked by recent studies,
* Presenting a novel contribution on finding the strong agreement-on-the-line (AGL) and accuracy-on-the-line (ACL) phenomena in TTA across various setups and baselines,
* Introducing straightforward yet highly effective methods for accuracy estimation, calibration, and hyperparameter optimization.

In our response to the reviewers, we will first address the concerns shared by multiple reviewers before individually addressing the specific concerns raised by each reviewer. The following section outlines the common concerns and our corresponding responses. **We revised our submission, and newly added writings / sections are colored in blue in our revised submission.**

---

>## The proposed method requires a full access to ID data (Reviewer a5t4, 72ab, gmmt).

To mitigate this concern, we investigated the feasibility and effectiveness of our methods under conditions where the access to ID samples is critically limited. Specifically, when applying our method to TENT on CIFAR10-C, CIFAR100-C, and ImageNet-C Gaussian Noise, we varied the ratio of ID samples from 100% (full) to 1%. We then iterated over multiple runs (i.e., 100 for CIFAR10 and 10 for ImageNet) with randomly selected subsets of ID and reported the mean / standard deviation of the accuracy estimation and the calibration results.

Across different setups, we found that **even given a significantly small number of ID samples (i.e., around 1-2% of total), our methods remain effective** in both estimating accuracy and calibration, comparable to the results assuming full access. The table below shows the mean absolute error (MAE) (\%) of accuracy estimation when using our methods with different setups, while varying the ratio of ID samples. We provided the CIFAR10-C results in appendix Section A.5 and Table 5 of the revised submission.

**ImageNet-C (MAE (\%))**
| Setup| 1%  | 2 % | 5 % | 10 % | 20 % | 50 % | 100 % |
| -|-|-|-|-|-|-|-|
| Architecture | 2.50±0.84 | 2.34±0.55 |2.32±0.41 | 2.37±0.26 | 2.32±0.18 |2.33±0.09 | 2.33 |
| Learning Rate | 4.52±0.71 | 4.60±0.47 |4.47±0.36 | 4.49±0.23 | 4.51±0.15 |4.51±0.08 | 4.51 |
| Adapt Step | 3.82±1.02 | 3.54±0.73 |3.54±0.38 | 3.55±0.34 | 3.50±0.20 |3.50±0.11 | 3.49 |
| Batch Size | 2.99±0.78 | 2.98±0.51 |3.02±0.30 | 3.01±0.23 | 3.04±0.14 |3.03±0.07 | 3.03 |
| Checkpoint | 2.09±0.66 | 1.71±0.46 |1.54±0.45 | 1.48±0.35 | 1.41±0.28 |1.43±0.15 | 1.43 |

In addition, we also present our method's calibration results given the same constraints, as shown in the table below. Here, we used different checkpoints, and we report the expected calibration error (ECE) (\%) results averaged over 15 corruptions on CIFAR100-C (See appendix Section A.5 and Table 6 for CIFAR10-C and ImageNet-C).

**CIFAR100-C (ECE (\%))**
| Setup | Uncalibrated | 1%  | 2 % | 5 % | 10 % | 20 % | 50 % | 100 % |
|-|-|-|-|-|-|-|-|-|
| Checkpoint | 13.40 | 3.61±1.4 |3.24±1.23 | 2.83±0.86 | 2.64±0.6 |2.52±0.44 |2.37±0.25|2.10|

Lastly, we would like to refer to the recent noteworthy studies of TTA that, akin to our approach, rely on the limited access to ID data, contributing to the mitigation of the catastrophic forgetting of ID performances [1] or enabling single-batch TTA [2]. Our work aligns with this line of research, aiming to enhance the reliability of TTA methods under conditions where limited access to in-domain data is feasible.

[1] Niu et al., Efficient Test-Time Model Adaptation without Forgetting, ICML’22

[2] Lim et al., TTN: A Domain-Shift Aware Batch Normalization in Test-Time Adaptation, ICLR’23

---

> ### Author Response · Authors · 2023-11-20
> **Response to All Reviewers [2/3]**
>
> >## Heavy reliance on multiple models (Reviewer mLJy, gmmt).
>
> It's worth noting that in practice, the validation in TTA involves running TTA iteratively with various hyperparameters, and selecting the one that yields the highest validation accuracy based on a labeled set. Our methods allow the same validation, which leverages multiple TTAed models and evaluate their performances, which might require a similar cost in memory in obtaining the final TTA model. Note that our methods do not require the labeled data for evaluation.
>
> Moreover, we have conducted the ablation study on the number of models in our original submission, which can be referred to Section A.4 and Figure 6 of the appendix (tested on Gaussian Noise of CIFAR10-C and ImageNet-C) in our revised submission.
>
> Here, we additionally provide the accuracy estimation results averaged over all corruptions in ImageNet-C as below. The tested setups are 1) architectures (2-4 columns), 2) learning rates (5-7 columns), and 3) batch sizes (8-10 columns). We showed that the MAEs when **using three (minimum) or four models, are competitive to those using the full available set of models** (denoted as full). These results indicate that in the memory-limited scenarios, practitioners can still robustly operate our methods with the minimal number of models to achieve reliable TTA performance predictions.
>
> **ImageNet-C (MAE (\%))**
>
> |Method | 3 | 4 | 9 (Full) | |  3 | 4 | 7 (Full) | |  3 | 4 | 8 (Full) |
> | ---------- | ---------- | ---------- | ---------- |  ---------- |  ---------- |  ---------- |  ---------- | ---------- |  ---------- |  ---------- |  ---------- |
> | ALine-S | 3.43±1.53 | 3.32±0.87 | 3.53±0.0 | | 5.65±3.07 | 4.60±1.20 | 4.41±0.0 | | 4.38±2.27 | 3.67±1.06 | 3.45±0.0 |
> | ALine-D | 3.53±1.64 | 3.38±0.90 | 3.47±0.0 | | 6.18±3.50 | 4.90±1.85 | 3.81±0.0 | | 4.13±2.61 | 3.04±1.31 | 2.36±0.0 |
>
> ---
> >## The proposed method works in offline manner (Reviewer a5t4, 72ab, gmmt).
>
> To address the concern, we validated the efficacy of our method in an online setup, where we dynamically conducted accuracy estimation and calibration on the data streamed in, as opposed to waiting for the completion of predictions on the entire test set (i.e., offline). Specifically, we calculated the AGL/ACL using the predictions of the current batch (or a few batches), and performed the real-time accuracy estimation and temperature-scaling of them.
>
> We examined our methods by using different setups on CIFAR10-C (averaged over every corruption), and employed only four models for memory efficiency. We tested 10 different combinations of models for each setup (4 for adapt step), and reported the mean and standard deviation of their results. In addition, to evaluate our method’s sensitivity to the size of data streamed, we systematically vary the number of batches to collect the predictions. For instance, if our method works only on the current batch, there is no delay (fully online), and if it waits until 2 batches of data are collected, there is 1 batch-delay.
>
> The table below demonstrates how **our methods, when applied an online setup, effectively perform accuracy estimation and calibration**. The gap between the fully online and offline version highlights that practitioners can utilize our methods in an online fashion to achieve low estimation error and ECE, comparable to those of offline version.
>
> **CIFAR10-C [MAE (\%)]**
> | Setups | 1 (Fully online) | 2 (1 Batch-delay) | 4 (3 Batch-delay) | Offline |
> | ---------- | ---------- | ---------- | ---------- | ---------- |
> | Architecture | 0.96±0.49 | 1.01±0.50 | 0.97±0.47 | 0.86±0.30 |
> | Learning rate | 2.50±1.72 | 2.28±1.48 | 2.18±1.45 | 2.65±1.67 |
> | Adapt step | 2.74±1.39 | 2.76±1.44 | 2.55±1.42 | 1.93±0.36 |
> | Checkpoint | 1.00±0.79 | 0.72±0.52 | 0.61±0.30 | 0.48±0.26 |
>
> **CIFAR10-C [ECE (\%)]**
> | Setups | Uncalibrated |1 (Fully online) | 2 (1 Batch-delay) | 4 (3 Batch-delay) | Offline |
> | ---------- | ---------- | ---------- | ---------- | ---------- | ---------- |
> | Architecture | 7.76 | 2.85±0.24 | 3.04±0.21 | 3.13±0.18| 3.18±0.10 |
> | Learning rate | 7.76 |  4.40±2.10 | 3.91±1.51 | 3.34±1.22 | 4.09±1.19 |
> | Adapt step | 7.76 | 4.33±1.96 | 4.08±1.75 | 4.03±1.94 | 3.37±0.15 |
> | Checkpoint | 7.76 |  2.53±0.57 | 2.69±0.36 | 2.89±0.19 | 3.13±0.08 |

---

> > ### Author Response · Authors · 2023-11-20
> > **Response to All Reviewers [3/3]**
> >
> > >## Empirical results on transformer-based models (Reviewer a5t4, 72ab, gmmt).
> >
> > We expanded our experiments to include transformer-based models, addressing the suggested extension of tested architectures. Since such models have layer normalization (LN) layers, we applied SAR [1] that is found effective in adapting models with LN.
> > We found that, after applying SAR[1], **such transformer-based models (with different architectures) are following the stronger AGL/ACL lines than vanilla** in ImageNet vs. ImageNet-C Gaussian Noise, very similar to those of CNN-based models (https://tinyurl.com/5n85czs4). At the same time, when tested on ImageNet-R (https://tinyurl.com/b87ffm45) and ImageNet-V2 (https://tinyurl.com/59wrxp96), they persist to have strong AGL/ACL, demonstrating that such phenomena are generalized to transformer-based models as well (here (second row)).
> >
> > For each figure, we marked “x” for the accuracy of transformers, while “o” for CNN-based models. We included 11 new models from three transformer-based backbones,
> > * ViT[2]: ViT-L/16, ViT-B/32,
> > * Swin Transformer [3]: Swin-T, Swin-S, Swin-B, and Swin-L
> > * Data-Efficient Image Transformer [4]: DeiT-S/16, DeiT-B/16, DeiT3-B/16, DeiT3-L/16, and DeiT3-H/14, from timm repository.
> >
> > We observed that transformers adapted with SAR show a high insensitivity to different adaptation hyperparameter values, unlike CNN-based models, particularly in their ID accuracy. This characteristic makes it less practical to define AGL/ACL using different hyperparameter values. However, our empirical findings regarding strong correlations across different architectures remain consistent for both CNN and transformer-based models. As mentioned in the discussion section, we posit that elucidating the precise conditions under which such linear correlations manifest remains an important and intriguing question for future research. We will incorporate discussions and these findings in our camera-ready version.
> >
> > [1] Niu et al., “Towards Stable Test-Time Adaptation in Dynamic Wild World”, ICLR’23
> >
> > [2] Dosovitskiy et al., “An image is worth 16x16 words: transformers for image recognition at scale”, ICLR’21
> >
> > [3] Liu et al., “Swin Transformer: Hierarchical Vision Transformer using Shifted Window”, ICCV’21
> >
> > [4] Touvron et al., “Training data-efficient image transformers & distillation through attention”, ICML’21
> >
> > ---
> > Up to this point, we have addressed general concerns, and we sincerely hope that our responses have sufficiently addressed the concerns.

---

### Meta-Review · Area_Chair_BvuX · 2023-12-06

**Metareview:**

This paper addresses the use of "test-time adaptation" to adapt models to distribution shifts in the test data.  As such shifts are commonplace in practice, and known to significantly degrade performance, better understanding and mitigating these seems important and valuable to the community.  An interesting insight from the authors is that many behaviors under test time distribution shift can be observed as having a linear trends.  The authors use this linear trend observation to devise mitigations that improve OOD performance.  According to review scores the paper is truly borderline, with two 5's and two 6's.  The reviewers all found the the key observation of the paper, the linear trend phenomenon, interesting and insightful.  They remarked that it could lead to valuable improvements in TTA to distribution shifts.  A key concern of the reviewers was that they found the experiments to be a little underwhelming, i.e. using just convolutional networks.  It seems like a reasonable concern that the observed phenomena could be particular to that architecture on vision problems.  Multiple reviewers asked for results on bigger or more modern models.  Multiple reviewers also flagged issues with comparisons to Baek et al., i.e. in terms of novelty and in discussion.

As stated above, the key insight is interesting and reviewers think it should be useful.  They were, however, underwhelmed by the experiments.  A sticking point is that the models used in the paper aren't really at or near the cutting edge of research. Multiple of the reviewers asked for comparisons to transformer-based models, which could help strengthen the claims of the work.  Unfortunately, no reviewers were willing to really champion the paper, and the scores were all borderline, rather than having high variance with a borderline average.  Therefore the recommendation is to reject unfortunately.  However, it seems that there is a strong start here for a future submission to another venue.  I.e. the reviewers all found the insights to be quite compelling, but were underwhelmed by experiments.  Hopefully the reviews will be helpful to strengthen the paper.

**Justification For Why Not Higher Score:**

None of the reviewers were willing to champion the paper.  It's ok, but underwhelming in the execution.  It could be made much stronger with some better experiments for a future submission.

**Justification For Why Not Lower Score:**

NA

---

### Decision · Program_Chairs · 2024-01-16

Reject